# Model results of OH airglow considering four different wavelength regions to derive night-time atomic oxygen and atomic hydrogen in the mesopause region

Tilo Fytterer[1], Christian von Savigny[2], Martin Mlynczak[3], Miriam Sinnhuber[1]

[1]Institute for Meteorology and Climate Research, Karlsruhe Institute of Technology, Eggenstein-Leopoldshafen, 76344, Germany

[2]Institute of Physics, University of Greifswald, Greifswald, 17489, Germany

[3]NASA, Langley Research Center, Hampton, Virginia, 23681-2199, USA

*Correspondence to:* Miriam Sinnhuber (miriam.sinnhuber@kit.edu)

**Abstract.** Based on the zero dimensional box model CAABA/MECCA-3.72f, an OH airglow model was developed to derive night-time number densities of atomic oxygen ($[O(^3P)]$) and atomic hydrogen ($[H]$) in the mesopause region (~75-100 km). The profiles of $[O(^3P)]$ and $[H]$ were calculated from TIMED/SABER satellite OH airglow emissions measured at 2.0 µm. The two target species were used to initialize the OH airglow model, which was empirically adjusted to fit four different OH airglow emissions observed by the satellite/instrument configuration TIMED/SABER at 2.0 µm and at 1.6 µm as well as measurements by ENVISAT/SCIAMACHY of the transitions OH(6-2) and OH(3-1). Comparisons between the "Best fit model" obtained here and the satellite measurements suggest that deactivation of vibrationally excited OH($v$) via OH($v \geq 7$)+$O_2$ might favour relaxation to OH($v' \leq 5$)+$O_2$ by multi-quantum quenching. It is further indicated that the deactivation pathway to OH($v'=v-5$)+$O_2$ dominates. The results also provide general support of the recently proposed mechanism OH($v$)+$O(^3P) \rightarrow$ OH($0 \leq v' \leq v-5$)+$O(^1D)$ but suggest slower rates of OH($v$=8,7,6,5)+$O(^3P)$, partly disagreeing with laboratory experiments. Additionally, deactivation to OH($v'=v-5$)+$O(^1D)$ might be preferred. The profiles of $[O(^3P)]$ and $[H]$ derived here are plausible between 80 km and 95 km but should be regarded as an upper limit. The values of $[O(^3P)]$ obtained in this study agree with the corresponding TIMED/SABER values between 80 km and 85 km, but are larger from 85 to 95 km due to different relaxation assumptions of OH($v$)+$O(^3P)$. The [H] profile found here is generally larger than

TIMED/SABER [H] by about 50 % from 80 to 95 km, which is primarily attributed to our faster OH($v$=8)+$O_2$ rate.

## 1 Introduction

Atomic oxygen in its ground state (O($^3$P)) and atomic hydrogen (H) strongly influence the energy budget in the mesopause region (~75-100 km) during day and night (Mlynczak and Solomo, 1993), and consequently affect atmospheric air temperature, wind, and wave propagation (Andrews et al., 1987). Therefore, an improved knowledge of the abundance of O($^3$P) and H is of great importance when studying the mesopause region. At these altitudes, O($^3$P) has a direct impact on the heating rates by

participating in several exothermic chemical reactions (Mlynczak and Solomon, 1993, their Table 4). But O($^3$P) also contributes to radiative cooling  by exciting $CO_2$ via collisions, leading to increased infrared emissions of $CO_2$ and partly opposing the O($^3$P) chemical heating effect. Night-time H plays a crucial role in the mesopause region due to the destruction of ozone ($O_3$) which is accompanied by the release of a considerable amount of heat (Mlynczak and Solomon, 1993). This chemical reaction

additionally leads to the production of vibrationally excited hydroxyl radicals (OH($v$>0)) up to the vibrational level $v$=9, causing the formation of OH emission layers in the atmosphere (Meinel bands; Meinel, 1950).

Direct measurements of O($^3$P) and H are relatively rare because as atomic species they do not have observable vibration-rotation spectra. Consequently, measuring these species in the mesopause region

by remote sensing requires complex methods while in situ observations are rather expensive (e.g. Mlynczak et al., 2004; Sharp and Kita, 1987). Thus, there exists no global data set based on direct observations. As a consequence, an indirect method was introduced by Good (1976) to derive [O($^3$P)] and [H] during night, using OH airglow emissions. This approach was also adapted by Mlynczak et al. (2013a; 2014; 2018) which derived a global data set of night-time [O($^3$P)] and [H] in the mesopause

region from satellite observations of OH($v$). The method is based on the assumption of chemical steady state of $O_3$ and further depends on several radiative lifetimes, chemical reactions, and physical processes involving OH($v$). However, the corresponding total rate coefficients and branching ratios are

still not sufficiently known, and thus present a large source of uncertainty in the derivation of [O($^3$P)] and [H].

There are two major issues currently discussed in the literature which considerably affect the overall abundance of derived O($^3$P) and H. The first problem addresses the underlying deactivation schemes of OH($v$) from the higher excited state $v$ to the lower excited state $v'$ ($v'<v$) by collisions with $O_2$. This can generally occur via sudden death (OH($v$)+$O_2$→OH($v'$=0)+$O_2$), single-quantum (OH($v$)+$O_2$→OH($v'$= $v$-1)+$O_2$), or multi-quantum (OH($v$)+$O_2$→OH($v'<v$)+$O_2$) quenching. However,

in case of the sudden death approach, it is still unknown where such a huge amount of excess energy is transferred. The second crucial point comprises the deactivation scheme and the total rate of OH($v$)+O($^3$P), including the new pathway OH($v$)+O($^3$P)→OH(0≤$v'$≤$v$-5)+O($^1$D) suggested by Sharma et al. (2015).

Over the last three to four decades, several model studies attempted to fit OH airglow measurements,

using different rates and schemes for the deactivation of OH($v$) by $O_2$ and by O($^3$P). And at least to our knowledge, there is no general agreement about which model is correct. The deactivation of OH($v$) by $O_2$ in many models (e.g. von Savigny et al., 2012; Mlynczak et al., 2013a; Grygalashvyly et al., 2014; Panka et al., 2017) is based on the model proposed by Adler-Golden (1997). It assumes a combination of multi-quantum and single-quantum quenching and was derived from theoretical considerations and

ground-based observations. Xu et al. (2012) investigated measurements from the Sounding of the Atmosphere using Broadband Emission Radiometry (SABER) instrument on board the NASA Thermosphere-Ionosphere-Mesosphere Energetics and Dynamics (TIMED) satellite of the OH airglow emissions at 2.0 µm and at 1.6 µm. Their results support the model of Adler-Golden (1997) but suggest slower total OH($v$)+$O_2$ rates. They further exclude the sudden death mechanism as a possible

deactivation scheme. There are also two theoretical studies (Shalashilin et al., 1995; Caridade et al., 2002) which investigated OH($v$) deactivation via $O_2$, both supporting a combination of multi-quantum and single-quantum quenching similar to the model of Adler-Golden (1997).

However, Russell and Lowe (2003) and Russell et al. (2005) analyzed OH(8-3) and O($^1$S) airglow emissions measured by the Wind Imaging Interferometer (WINDII) instrument on board the Upper

Atmospheric Research Satellite (UARS). Both airglow emissions were used to derive separate data sets

of [O($^3$P)] and the best agreement between these two [O($^3$P)] data sets was obtained when a sudden death scheme for OH(v)+O$_2$ quenching was applied. Kaufmann et al. (2008) investigated several OH airglow spectra between 1 µm and 1.75 µm measured by the Scanning Imaging Absorption Spectrometer for Atmospheric Chartography (SCIAMACHY) instrument on board the Environmental Satellite (ENVISAT). They found best agreement between their model and the measured OH airglow spectra when a combination of sudden death and single-quantum quenching was used.

Vibrationally dependent rates of OH(v)+O($^3$P) were determined by Varandas (2004) and Caridade et al. (2013), using quasi-classical trajectory calculations. Their results suggest that deactivation occurs via a chemical reaction as well as multi-quantum quenching. Kalogerakis et al. (2011) obtained a deactivation rate of OH(v=9)+O($^3$P) from laboratory experiments which is several times larger than the rate from these calculations. But applying this fast quenching rate led to non-physical [O($^3$P)] values and associated heating rates (Smith et al., 2010; Mlynczak et al., 2013a). Thus, Sharma et al. (2015) proposed a new mechanism OH(v)+O($^3$P)→OH(0≤v'≤v-5)+O($^1$D) to account for results from both theory and experiment. Very recent laser experiments and model studies support this new pathway while the exact values of the branching ratios and total loss rates are still not known (Kalogerakis et al., 2016; Panka et al., 2017). However, recently published results by Mlynczak et al. (2018) oppose this mechanism. They also applied the new rate of Kalogerakis et al. (2011) for OH(v=9)+O($^3$P). But in order to get the annual energy budget into near balance, it was necessary to assume that at least OH(v=9)+O($^3$P) occurs via single-quantum relaxation. Additionally, the rate of OH(v=8)+O$_2$ had to be reduced and is considerably smaller than the value reported from Adler-Golden (1997).

The newly suggested rates of OH(v)+O($^3$P) were applied in different models to derive [O($^3$P)] in the mesopause region. Mlynczak et al. (2018) used SABER OH airglow emissions observed at 2.0 µm to derive [O($^3$P)] and assumed rates of $3.0\times10^{-10}$ cm$^3$ s$^{-1}$ and $1.5\times10^{-10}$ cm$^3$ s$^{-1}$ for OH(v=9)+O($^3$P) and OH(v=8)+O($^3$P), respectively. They further stated that deactivation of OH(v=9)+O($^3$P) has to occur via single-quantum quenching and that the OH(v=8)+O$_2$ rate has to be smaller than known from laboratory measurements to get global annual energy budget into near balance. Panka et al. (2018) simultaneously investigated SABER OH airglow emissions measured at 2.0 µm and 1.6 µm, while applying faster rates

for OH($\nu$=8)+O($^3$P) and OH($\nu$)+O$_2$. Their [O($^3$P)] values agree within the corresponding errors with the results reported by Mlynczak et al. (2018) above ~87 km but are larger in the altitude region below. The authors also demonstrated the high sensitivity of the derived [O($^3$P)] from O($^3$P) quenching rates applied in their model. Zhu and Kaufmann (2018) analyzed SCIAMACHY OH(9-6) transition. They used a value of $2.3\times10^{-10}$ cm$^3$ s$^{-1}$ for OH($\nu$=9)+O($^3$P) which is lower than the one applied in the two previous studies, resulting in generally lower [O($^3$P)] values in the altitude region above 87 km. Their rate for OH($\nu$=9)+O$_2$ lies between the corresponding rates of the two other studies, and consequently their [O($^3$P)] is also between the [O($^3$P)] values of these two studies below 87 km. Thus, recent publications indicate that the rate of OH($\nu$=9,8)+O($^3$P) might be slower than previously suggested in Sharma et al. (2015). But this problem needs further attention because all three papers derive different [O($^3$P)], depending on the data sets investigated.

In order to address the two major issues stated above, this paper is focused on the development of a zero dimensional box model for atmospheric OH airglow with the intention to derive night-time [O($^3$P)] and [H] in the mesopause region. The model considers the formation of OH($\nu$) via H+O$_3$ and deactivation of OH($\nu$) due to spontaneous emission of photons, chemical reactions and physical collisions with atmospheric air compounds N$_2$, O$_2$, and O($^3$P). We used the indirect method introduced by Good (1976) and derived night-time [O($^3$P)] and [H] from TIMED/SABER OH emissions at ~2.0 µm, while also considering the OH airglow observations from TIMED/SABER at ~1.6 µm as well as the OH(6-2) and OH(3-1) transitions measured by ENVISAT/SCIAMACHY. Further sensitivity runs were carried out to estimate the uncertainty on the derived values of [O($^3$P)] and [H] due to the different deactivation schemes, overall rate constants, and branching ratios.

## 2 Data and method

### 2.1 Satellite measurements

#### 2.1.1 ENVISAT/SCIAMACHY

The SCIAMCHY instrument (Bovensmann et al., 1999) was an 8-channel spectrometer on board

ENVISAT, providing atmospheric OH airglow emission measurements between ~220 nm and ~2380 nm. ENVISAT was launched into a polar and sun-synchronous orbit and crossed the equator at ~10 LT

and ~22 LT. The ENVISAT mission started in March 2002 and SCIAMACHY was nearly continuously operating until the end of the mission in April 2012 caused by a spacecraft failure. The SCIAMACHY instrument performed measurements in different observations modes, including night-time (~22 LT) limb scans over the tangent altitude range ~75-150 km. These measurements are only available throughout the year at latitudes between the equator and 30° N.

In this paper, we used SCIAMACHY level 1b data v7.04 to retrieve OH airglow volume emission rates (VERs) of the OH(3-1) and OH(6-2) bands in the wavelength ranges of 1515-1546 nm and 837.5-848.0 nm, respectively. The retrieval approach applied here is very similar to the one described in von Savigny et al. (2012). The retrieval does not cover the complete spectra of the OH(3-1) and OH(6-2) bands, and consequently a "correction factor" of 2.48 for OH(3-1) VER and 2.54 for OH(6-2) VER was applied to

account for the entire band emissions at mesopause temperature. The data set further includes corrections for misalignments and other measurement errors (Gottwald et al., 2007). Investigations performed by Bramstedt et al. (2012) showed a drift of the SCIAMACHY tangent height of less than 20 m year$^{-1}$ which is negligible for our study.

The uncertainties of the OH(3-1) VER and OH(6-2) VER retrievals from SCIAMACHY limb

observations correspond to the propagated uncertainties of the observed limb emission rate (LER) profiles. The latter are estimated from the LER values in the tangent height range between 110 km and 150 km, where the actual atmospheric emissions should be zero. The VER uncertainties are first determined for daily and zonally averaged data. The uncertainties used in this analysis correspond to the mean uncertainties averaged over all days with co-located SCIAMACHY and SABER observations.

**2.1.2 TIMED/SABER**

The SABER instrument (Russell et al., 1999) on board the TIMED satellite has been nearly continuously operating since January 2002, collecting over 98 % of all possible data. The instrument scans the atmosphere from the surface up to altitudes of ~400 km while providing a vertical resolution of about 2 km throughout the entire height interval. Due to the geometry of the satellite orbit and the

regular yaw manoeuvres every ~60-65 days, SABER only provides complete coverage of the latitude range between ~55° S and ~55° N. The SABER instrument measures the OH VERs at ~2.0 µm and at ~1.6 µm which approximately corresponds to the transitions of OH(9-7)+OH(8-6) and OH(5-3)+OH(4-2), respectively. The contribution of OH(7-5) to OH VER at 2.0 µm and of OH(3-1) to OH VER at 1.6 µm is only about a few percents (Xu et al., 2012; Mlynczak et al., 2013a) and is neglected in this paper.

In this study, we used the SABER Level 2A data v2.0 of the "unfiltered" OH VERs at 2.0 µm and at 1.6 µm, the air temperature and pressure, and the volume mixing ratios (VMRs) of $O_3$ (derived at 9.6 µm). There are also SABER $O_3$ measurements at 1.27 µm but these observations are not available during night. New night-time VMRs of $O(^3P)$ and H (Mlynczak et al., 2018) were used for comparison with the results derived from our model. The "unfilter" factor applied to OH VER adjusts the originally

measured OH VER by the SABER instrument to the total VER emitted by OH in the corresponding vibrational bands, while considering the shape, width, and transmission of the SABER broadband filters (Mlynczak et al., 2005). Outliers were excluded by screening the data as suggested by Mlynczak et al. (2013a). The SABER data used here were further restricted to observations between 21 LT and 23 LT to approximately match the SCIAMACHY measurement time at ~22 LT. In order to be consistent with the

naming of the SCIAMACHY OH airglow observations, the SABER OH airglow at 2.0 µm and at 1.6 µm are referred to as OH(9-7)+OH(8-6) and as OH(5-3)+OH(4-2) throughout the paper.

The total uncertainty of SABER OH airglow data used here comprises three different error sources. Since we used climatology of the measurements (see Sect. 2.2), there are sufficient samples that the random noise component of the total uncertainty is essentially zero. The remaining two major terms are

the absolute calibration error (<5 %) and the "unfilter" factor error (<3 %). Assuming a root-sum-square propagation of the individual uncertainties, this results in a total uncertainty of about 6 % for all data points presented in this study.

## 2.2 Method

In order to minimize uncertainties between SABER and SCIAMACHY due to different measurement
characteristics, we focused on the latitude range from 0° to 10° N, which was covered by both instruments throughout the entire year. A broader latitude band is not recommended because SABER

and SCIAMACHY do not uniformly cover the same latitudes, leading to disagreements between the real latitude of the observations and the nominal latitude of the interval. The accepted profiles of both instruments within the chosen latitude interval were averaged to zonal mean nightly mean values. All these zonal mean nightly means from January 2003 to December 2011 were used to calculate a climatology, including only days on which both SCIAMACHY and SABER data are available.

The approach to derive $[O(^3P)]$ and $[H]$ applied here was developed by Good (1976) and is described in detail in Mlynczak et al. (2013a). Thus, we only give a brief summary here. The measured SABER OH(9-7)+OH(8-6) VER (photons cm$^{-3}$ s$^{-1}$) is given by Eq. (1):

$$OH(9\text{-}7) + OH(8\text{-}6)\,VER = k_1[H][O_3]G(f_v, A_{vv'}, C_{vv'}), \qquad (1)$$

where $k_1$ is the rate constant of the chemical reaction $H+O_3$, representing direct production. The function $G$ (Eq. (2)) comprises all relevant production and loss processes of OH(9-7) VER and OH(8-6) VER:

$$G = \frac{f_9}{A_9 + C_9}A_{97} + \frac{f_8}{A_8 + C_8}A_{86} + \frac{f_9}{A_9 + C_9}\frac{A_{98} + C_{98}}{A_8 + C_8}A_{86}, \qquad (2)$$

The subscripts $v$ and $v'$ ($v'<v$) are the vibrational states of OH before and after the corresponding process. The terms $f_v$ are the nascent distributions and describe the production efficiency of OH($v$) via the reaction $H+O_3$. Total radiative loss due to spontaneous emissions is considered by the Einstein coefficients $A_v$ (s$^{-1}$) which are the inverse radiative life times of OH($v$). The total loss rate $C_v$ (s$^{-1}$) is the sum of loss due to collisions with the air compounds ($N_2$, $O_2$, $O(^3P)$), including chemical reactions and physical quenching. The terms $A_{vv'}$ and $C_{vv'}$ represent the specific state-to-state transitions.

In the second step, chemical equilibrium of $O_3$ during night is assumed as follows:

$$k_1[H][O_3] + k_2[O(^3P)][O_3] = k_3[O(^3P)][O_2][M], \qquad (3)$$

meaning that $O_3$ loss due to H and $O(^3P)$ (left side) is balanced by $O_3$ formation via the three-body-reaction $O(^3P)+O_2+M$ (right side). Here, $k_2$ and $k_3$ are the corresponding rate constants of $O(^3P)+O_3$ and $O(^3P)+O_2+M$, respectively, while $M$ being an air molecule and $[M]$ being the total number density of the air.

Finally, rewriting Eq. (1) enables the derivation of $[H]$ while $[O(^3P)]$ is calculated by substituting Eq. (3)

in Eq. (1) and rewriting the resulting term as follows:

$$[H] = \frac{OH(9-7)+OH(8-6)\ VER}{G\,k_1\,[O_3]}, \qquad (4a)$$

$$[O(^3P)] = \frac{OH(9-7)+OH(8-6)\ VER}{G\,(k_3\,[O_2][M]-k_2\,[O_3])}. \qquad (4b)$$

Air temperature and air pressure from SABER were used to calculate [M], [$O_2$] (VMR of 0.21), and [$N_2$] (VMR of 0.78) via the ideal gas law, and [M] was then used to convert SABER $O_3$ VMR into [$O_3$]. The chemical reaction rates and physical quenching processes involved are described in Sect. 2.3. The values of [$O(^3P)$] and [H] were individually derived for each altitude. Finally, the obtained vertical profiles of [$O(^3P)$] and [H] were used to initialize the OH airglow model (see Sect. 2.3).

It is apparent from Eq. (4a-b) that any changes applied to the input parameters (G, $O_2$, $O_3$, M, $k_1$, $k_2$, $k_3$) are balanced by the derived values of [$O(^3P)$] and [H], without assuming any a priori information of [$O(^3P)$) and [H]. In contrast, OH(9-7)+OH(8-6) VER is not affected by the input parameters and therefore identical in every model run. However, the goal of this paper is to develop a model which does not only fit OH(9-7)+OH(8-6) VER observations but also reproduces the three other airglow measurements OH(6-2) VER, OH(5-3)+OH(4-2) VER, and OH(3-1) VER. We have to further point out, that the relation between [$O(^3P)$] and OH(9-7)+OH(8-6) VER is not linear since the function G also depends on [$O(^3P)$], as represented by the terms $C_v$ and $C_{vv'}$. In fact, Eq. (4b) is a quadratic expression with respect to [$O(^3P)$] but treated here as a linear one, making no substantial differences for small [$O(^3P)$]. Nevertheless, this issue is addressed in detail in Sect. 3.4.

## 2.3 The OH airglow Base model

The model used in this study is based on the atmospheric chemistry box model Module Efficiently Calculating the Chemistry of the Atmosphere/Chemistry As A Box model Application (MECCA/CAABA-3.72f; Sander et al., 2011). The box model calculates the temporal evolution of chemical species inside a single air parcel of a certain pressure and temperature, making the model well suited for sensitivity studies. The CAABA/MECCA standard model was extended by several chemical reactions and physical quenching processes involving OH(v) which are described in this section. The

model was run until it reaches steady-state, defined by the agreement between the measured and modelled OH(9-7)+OH(8-6) VER.

The OH airglow model described in this section is referred to as "Base model" because it is the starting point of our model studies. But we have to point out that there is no such a thing as a commonly accepted OH airglow base model in the literature. The Base model takes into account all major formation and loss processes of $OH(v)$ (Table 1) which are commonly used in other models in the literature and are assumed not to be seriously in error. The model comprises the production of $OH(v)$

via the chemical reaction $H+O_3$ as well as the deactivation due to spontaneous emission and the removal physical quenching and chemical reactions with $N_2$, $O_2$, and $O(^3P)$.

The chemical reactions $H+O_3$, $O(^3P)+O_3$, and $O(^3P)+O_2+M$ were already included in the CAABA/MECCA standard model and their corresponding rates were taken from the latest Jet Propulsion Laboratory (JPL) report 18 (Burkholder et al., 2015). The reaction $H+O_3$ can populate

$OH(v)$ at all vibrational levels $v \leq 9$ and the nascent distribution of $OH(v)$ was taken from Adler-Golden (1997). The spontaneous emissions are given by the Einstein coefficients at 200 K (Xu et al., 2012). Deactivation of $OH(v)$ by $N_2$ is assumed to occur via single-quantum quenching. The rates at room temperature for $OH(v \leq 8)$ and for $OH(v=9)$ were taken from Adler-Golden (1997) and Kalogerakis et al. (2011), respectively.

Quenching of $OH(v)$ by $O_2$ is based on the values reported by Adler-Golden (1997, their Table 3) which comprise a combination of multi-quantum and single-quantum quenching. However, Adler-Golden (1997) applied a factor of ~1.5 to account for mesopause temperature based on comparisons between laboratory measurements at room temperature of $OH(v=8)+O_2$ and the corresponding rate inferred from OH(8-3) rocket observations in the mesopause region. But later experiments reported by Lacousiere et

al. (2003) and calculations by Caridade et al. (2002) suggest smaller values. The latter study further indicates that the temperature dependence decreases for lower vibrational levels and becomes negligible for $OH(v \leq 4)$. Consequently, the rates presented in Adler-Golden (1997) were scaled to room temperature measurements ($v$=1-6 Dodd et al., 1991; $v$=7 Knutsen et al., 1996; $v$=8 Dyer et al., 1997; $v$=9 Kalogerakis et al., 2011), and afterwards a factor of 1.1 for $OH(v \geq 6)$ and 1.05 for $OH(v=5)$ was

applied.

The removal of OH(v) via collisions with O($^3$P) is included by using a combination of multi-quantum quenching (Caridade et al., 2013, their Table 1) and chemical reactions (Varandas, 2004). The rates were obtained from quasi-classical trajectory calculations at 210 K, approximately matching mesopause temperature.

As described in the previous section, the OH airglow model is adjusted to fit OH(9-7)+OH(8-6) VER, OH(6-2) VER, OH(5-3)+OH(4-2) VER, and OH(3-1) VER. Thus, the model cannot provide information about OH(v≤2). It further treats OH(v=9) and OH(v=8) as well as OH(v=5) and OH(v=4) as a single level and the corresponding deactivation channels presented in Table 2 and 3 should be viewed more critically.

## 3 Results and discussions

Figure 1 displays vertical profiles of a) OH(6-2) VER, b) OH(5-3)+OH(4-2) VER, and c) OH(3-1) VER, comparing the observations (black squares) with the corresponding Base model output (red line). The model results of OH(6-2) VER and OH(3-1) VER are a 4 km running average to take the averaging kernels of SCIAMACHY measurements into account. The Base model approximately matches the
general shape of the measured profiles but overestimates the three OH airglow measurements at the altitude of maximum VER. A closer look at the relative differences shows that the ratio model/observation at the altitude of maximum VER is about 2.0, 1.2, and 1.3 for OH(6-2), OH(5-3)+OH(4-2), and OH(3-1), respectively. Furthermore, these ratios increase with decreasing altitude, indicating that the overestimation of the Base model might be associated with $O_2$ quenching.

The differences between Base model and observations are quite substantial in case of OH(6-2) VER. This implies a general problem of the rates or schemes included in the Base model, requiring a detailed error analysis. The focus was set on potential error sources of OH(6-2) VER because the relative differences between model and measurements are largest compared to the other two OH transitions, and secondly because changes of OH(v=6) will affect the lower vibrational levels, but not vice versa.

**3.1 Potential error sources of OH(6-2) VER in the Base model**

Based on the results presented in Fig. 1, the potential error source has to have an effect on the entire height interval and must have a stronger impact on OH(6-2) compared to the other two OH transitions. We further focus on quantities with large uncertainties. For the latter reason, temperature is excluded as possible source because to account for a reduction of OH(6-2) VER by a factor of 2, temperature must be increased by more than 20 K (not shown here). Such a large error is very unlikely considering that a zonal mean climatology (2003-2011) is used here.

Since the overestimation of the Base model is especially large for OH(6-2) VER, an impact of the Einstein coefficient of the corresponding transition must be considered. Regarding this aspect, we have to point out that studies based on HITRAN 2004 data set should be viewed more critically, because of erroneous OH transition probabilities. The Einstein coefficients used in this study were recently recalculated (Xu et al., 2012, their Table A1) and correspond to a temperature of 200 K, which is very close to mesopause temperature. Furthermore, these Einstein coefficients are consistent with the values of the HITRAN 2008 data set (Rothman et al., 2009). However, there are several other data sets of Einstein coefficients found in literature that might lead to different results. We therefore carried out sensitivity runs, using the Einstein coefficients reported by Turnbull and Lowe (1989), Nelson et al. (1990), van der Loo and Groenenboom (2007), Xu et al. (2012; =Base model), and Brooke et al. (2016). The corresponding results are presented in Figure 2 and show considerably large differences in case of OH(6-2) VER which are about a factor of 4 between the highest and lowest model output. In contrast, the individual simulations of OH(5-3)+OH(4-2) VER and OH(3-1) VER are rather consistent and vary only by ~10 %. These results emphasize that the choice of the Einstein coefficients is a potential error source for higher quanta transitions.

Regarding the credibility of the Einstein coefficients, it is generally assumed that the calculation improve with time. However, this is not necessarily true at quanta changes >2 because it all depends on how good the representation of the Hamiltonian for the OH molecule is, that is used to solve the Schrödinger equation. Multi quanta transitions >2 quanta have small Einstein coefficients and are generally hard to model and calculate. The assessment of the Einstein coefficients requires a detailed analysis of the corresponding calculations, which is beyond the scope of this study. We therefore cannot exclude the values used in the Base model as a potential error source, but we also think that our choice

of the Einstein coefficients from Xu et al. (2012) is reasonable. Additionally, these values represent approximately the average model output of all five data sets considered here, while the model results based on Nelson et al. (1990) and van der Loo and Groenenboom (2007) represent the variability. Thus, we will not replace the Einstein coefficients by Xu et al. (2012) in our model but keep in mind that they might be too large.

Furthermore, the best agreement between the observations and the model was obtained by applying the Einstein coefficients reported by van der Loo and Groenenboom (2007). But even in this case, the model still overestimates the observations of all OH transitions in the altitude region between ~80 km and ~86 km. This pattern strongly supports the suggestion stated above that the rates and schemes associated with OH(v)+$O_2$ are incorrect.

The nascent distribution of the excited OH states of the chemical reaction H+$O_3$ was observed in several studies and all of them agree that OH(v) is primarily formed in the vibrational levels v=8 and v=9 (e.g. Charters et al., 1971; Streit and Johnston, 1976; Ohoyama et al., 1985; Klenerman and Smith, 1987). The values used in the Base model were taken from Adler-Golden (1997) which are based on measurements reported by Charters et al. (1971) and agree with values obtained by Klenerman and Smith (1987) and Streit and Johnston (1976). The values found by Ohoyama et al. (1985) show some differences, but according to Klenerman and Smith (1987), their results are fundamentally flawed. This also affects the nascent distribution used by Mlynczak and Solomon (1993) which is an average of Charters et al. (1971), Ohoyama et al. (1985), and Klenerman and Smith (1987).

Therefore, we think that our nascent distribution used here is likely not a serious error source. However, minor errors might be introduced by extrapolating the nascent distribution to lower vibrational levels as it was done for the values used in our study (Adler-Golden, 1997). It is also possible that part of the nascent value of OH(v=6) is not due to direct production via H+$O_3$ but results from contributions of OH(v≥7). In order to test the potential impact of the OH(v=6) nascent value on OH(6-2) VER, we assumed an extreme scenario by reducing the OH(v=6) nascent value from 0.03 to zero. But the corresponding results of OH(6-2) VER of the Base model run (not shown here) are only about 15 % lower compared to the values presented in Fig. 1. Further sensitivity runs also showed that an increase of the ratio $f_9/f_8$ is associated with a decrease of modelled OH(6-2) VER but even the extreme case of

$f_9=1$ and $f_8=0$ could not account for a factor of 2. Note that changes of the overall rate constant of $H+O_3$ affect all considered OH transitions in a similar way, Thus, we conclude that direct production of $OH(v)$ is unlikely to be the reason for the overestimation of OH(6-2) VER by the Base model.

The physical removal of $OH(v)$ by $N_2$ is included as single-quantum relaxation which is supported by theoretical studies (Shalashilin et al., 1992; Adler-Golden, 1997). Assuming a sudden death scheme with the same overall deactivation rates resulted in a decrease of simulated OH(6-2) VER by less than 10 % at the altitude of maximum VER. The total deactivation rate for $OH(v=9)$ used here is about 1.5 times higher than the one suggested by Adler-Golden (1997) but the difference between the corresponding

model OH(6-2) VERs is negligible (<1 %). There are two studies reporting temperature dependence of $N_2$ quenching (Shalashilin et al., 1992; Burtt and Sharma, 2008), both agreeing with measurements at room temperature. However, the calculations of the former study imply slower quenching rates at mesopause temperature compared to their respective values at room temperature whereas the latter publication indicates the opposite behaviour, reporting a ratio between the rate at 200 K and 300 K of

approximately 1.7 for $OH(v=8)$ and 1.3 for $OH(v=9)$. These factors are generally supported by López-Puertas et al. (2004) which applied an empirically determined factor of 1.4 to the rates of Adler-Golden (1997) to account for mesopause temperature. Since the temperature dependence is still uncertain, we tested both possibilities. We increased and decreased the overall $OH(v)+N_2$ quenching rates by a factor of 1.5 which led to higher or lower OH(6-2) VERs by about 5 %. Therefore, $N_2$ is too inefficient as a

$OH(v)$ quenching partner to cause differences of OH(6-2) VER of a factor of 2.

    The overall rate and exact pathways of $OH(v)+O(^3P)$ are also still not known well enough but $O(^3P)$ has nearly no influence on $OH(v)$ at altitudes below 85 km. It therefore cannot be the only reason for the differences presented in Fig. 1. Consequently, deactivation by $O_2$ is the only remaining candidate which has a crucial influence on $OH(v)$ throughout the entire height interval. Therefore, we will first focus on

$OH(v)+O_2$ (Sect. 3.2) before investigating a potential influence of $O(^3P)$ on $OH(v)$ in Sect. 3.3.

### 3.2 Deactivation of $OH(v)$ by $O_2$

The overestimation of OH(6-2) VER by the Base model can be generally corrected either by slower rates of $OH(v=9,8,7)+O_2$ or by a faster rate of $OH(v=6)+O_2$. The overall deactivation of $OH(v=9)$ was

measured by Chalamala and Copeland (1993) and they recommended a value of $2.1 \times 10^{-11}$ cm$^3$ s$^{-1}$. This result was later confirmed by Kalogerakis et al. (2011), reporting a rate of $2.2 \times 10^{-11}$ cm$^3$ s$^{-1}$. The rates for OH($\nu$=8,7,6)+O$_2$ are each based on a single study only ($\nu$=8 Dyer et al., 1997; $\nu$=7 Knutsen et al., 1996; $\nu$=6 Dodd et al., 1991). But at least to our knowledge, there are no signs that the rates of OH($\nu$=9,8,7,6)+O$_2$ are fundamentally flawed. In order to test the impact of the individual rates on OH(6-2) VER, we carried out sensitivity runs by varying the overall rates within their recommended 2$\sigma$ errors. Thus, we reduced the values of OH($\nu$=9,8,7)+O$_2$ to $16 \times 10^{-12}$ cm$^3$ s$^{-1}$, $7 \times 10^{-12}$ cm$^3$ s$^{-1}$, and $5 \times 10^{-12}$ cm$^3$ s$^{-1}$, respectively, while the rate of OH($\nu$=6)+O$_2$ was increased to $4.5 \times 10^{-12}$ cm$^3$ s$^{-1}$. But even under this favoured condition, the Base model output of OH(6-2) VER decreased only by a factor of 1.5, still not close to the required difference of a factor of 2. Additionally, the assumed scenario is rather unlikely since the overall rates were obtained by independent studies.

The possibility of a systematic offset of OH($\nu \leq$6)+O$_2$ rates, which are based on the single study (Dodd et al., 1991), is also excluded because of the very good agreement of this OH($\nu$=2)+O$_2$ rate with the value obtained by Rensberger et al. (1989). Furthermore, when we increased the OH($\nu \leq$6)+O$_2$ rates by a factor of 3, the Base model approximately fits OH(6-2) VER and OH(3-1) VER but underestimates OH(5-3)+OH(4-2) VER by more than 30 %. Temperature dependence also affects the O$_2$ deactivation rates used here. But the factor to account for mesopause region temperature is suggested to be lower than 1.3 (Lacousiere et al., 2003; Cadidade et al., 2002) which has a weaker impact on OH(6-2) VER than the scenarios considered above.

Consequently, when applying the standard deactivations rates and schemes found in the literature, neither errors of the overall rates nor uncertainties of the temperature dependence can give a reasonable explanation of the overestimation of OH(6-2) VER Base model output shown in Fig. 1a. Since the overall rates were actually measured while the deactivation schemes are solely based on theoretical considerations, it is more convincing that the potential error source lies within OH($\nu$)+O$_2$ deactivation scheme rather than in the deactivation rates.

In order to considerably reduce OH(6-2) VER, we assumed an extreme scenario and substituted the multi-quantum relaxation (OH($\nu$)+O$_2$→OH($\nu$'<$\nu$)+O$_2$) in the Base model by a sudden death

(OH(ν)+O$_2$→OH+O$_2$) approach. This new model is referred to as "O$_2$ SD model" and the corresponding results are displayed in Fig. 3 as red lines, showing that the simulated OH(6-2) VER matches the observations within the error bars below 85 km and above ~92 km. The model still overestimates the measurements in the altitude region ~90 km, which might be related to O($^3$P)

quenching (see Sect. 3.3). The O$_2$ SD model output for the other two OH transitions (Fig. 3b-c) is clearly too low, implying that OH(ν)+O$_2$ quenching cannot occur via sudden death alone. We also conclude that the contribution of higher excited states OH(ν≥7) to OH(ν=6) must be negligible or even zero and these higher states are suggested to primarily populate lower vibrational levels OH(ν≤5). Therefore, OH(ν)+O$_2$ has to occur via multi-quantum quenching because in case of single-quantum

deactivation the contribution of OH(ν≥7) to OH(ν=6) is considerably larger than zero.

According to Finlayson-Pitts and Kleindienst (1981), OH(ν) might be relaxing to ν'=ν-5 while the excess energy is transferred to form O$_2$(b$^1$Σ). This vibration-to-electronic energy transfer was also mentioned by Anlauf et al. (1968) and is supported by the close energy match of the transition from OH(ν=9) to OH(ν=4) and from O$_2$(X$^3$Σ) to O$_2$(b$^1$Σ) of about 36.6 kcal mol$^{-1}$ and 37.5 kcal mol$^{-1}$,

respectively. Although there is no experimental support of this deactivation pathway, this approach gives a reasonable explanation for the observed pattern in our study and OH(ν) as a potential source of excited O$_2$, as discussed in Howell et al. (1990) and Murtagh et al. (1990). However, evaluating whether the product is really O$_2$(b$^1$Σ) or another excited O$_2$ state is beyond the scope of this study. Thus, we concluded that deactivation of OH(ν) by O$_2$ has to satisfy the following condition:

OH(ν≥6)+O$_2$ → OH(ν'≤5)+O$_2$        (R8)

while we further assume that the pathway

OH(ν≥6)+O$_2$ → OH(ν'=ν-5)+O$_2$          (R9)

is the preferred deactivation channel.

In order to test whether R9 could be the only pathway of R8 we assumed multi-quantum relaxation via:

OH(ν)+O$_2$ → OH(ν-5)+O$_2$              (R10a)

or OH(ν)+O$_2$ → OH(ν-4)+O$_2$              (R10b).

If R10a is integrated in the model (Fig. 3b-c, O$_2$ ν-5 model), the corresponding model output at altitudes

<90 km is only about 10 % below the observations of OH(5-3)+OH(4-2) VER and approximately matches OH(3-1) VER measurements within the error bars. The underestimation of the OH(5-3)+OH(4-2) VER measurements by the model could be attributed to minor errors of the OH(ν)+O$_2$ overall rates in combination with a slightly different OH(ν) branching of H+O$_3$. Therefore, we cannot completely rule out R10a as a possible solution, even if there are still some differences between the modelled and the observed OH VER. Replacing R10a by R10b in the model (Fig. 3b-c, O$_2$ ν-4 model) results in an overestimation of the observations of OH(5-3)+OH(4-2) VER and OH(3-1) VER by about 20 % to 30 %, and consequently this assumption is not further considered as a potential solution.

The results shown in Fig. 3 suggest that the OH airglow model is not able to reproduce the three OH airglow observations when sudden death or simplified multi-quantum schemes for OH(ν)+O$_2$ are applied. But the O$_2$ ν-5 model output is quite close to the measurements, suggesting that R9 might be the dominating deactivation channel within a multi-quantum relaxation scheme in accordance with R8. We therefore included these two conditions in the so-called "O$_2$ best fit model" and the results are displayed in Fig. 4. The corresponding branching ratios for the individual pathways are summarized in Table 2.

The simulated OH airglow fits well with the three OH airglow observations within the error bars below 85 km. In the altitude region above 85 km, it is seen that the model still overestimates OH(6-2) VER while OH(3-1) VER is indicated to be slightly underestimated. Furthermore, this pattern is not seen in OH(5-3)+OH(4-2) VER and therefore could be attributed to deviations due to the different satellite/instrument configurations between TIMED/SABER and ENVISAT/SCIAMACHY. But since this behaviour only occurs in the upper part of the vertical profiles and is not seen throughout the entire height interval, it is more likely related to O($^3$P) quenching.

**3.3 Deactivation of OH(ν) by O($^3$P)**

Only recently, Sharma et al. (2015) proposed a new pathway of OH(ν)+O($^3$P) by providing a direct link between higher and lower vibrational levels via:

$$OH(ν)+O(^3P) \rightarrow OH(0 \leq ν' \leq ν-5)+O(^1D) \qquad (R11),$$

with the vibrationally independent reaction constant $k_{11} = 2.3 \times 10^{-10}$ cm$^3$ s$^{-1}$. While the value of $k_{11}$(ν=9)

is based on measurements (Kalogerakis et al., 2011; Thiebaud et al., 2010) and on calculations (Varandas, 2004), the values for $k_{11}(v=5, 6, 7, 8)$ are only assumed to be identical to $k_{11}(v=9)$ and should be viewed more critically.

We adapted R11 in the "$O_2$ best fit $O(^3P)$ v-5 model" in such a way that the product is $OH(v'=v-5)+O(^1D)$ and the results obtained are displayed as blue lines in Fig. 5. Comparisons for OH(6-2) VER

in Fig. 5a show an underestimation of the model at altitudes >85 km. A sensitivity study was carried out that showed that the impact of $OH(v=9,8,7)+O(^3P)$ on OH(6-2) VER is negligible. This is reasonable because these three upper states only indirectly influence OH(6-2) via R11. Consequently, our analysis suggests a lower value of $k_{11}(v=6)$ and best agreement between model output and OH(6-2) VER observations was obtained for an overall rate of approximately $0.8\times10^{-10}$ $cm^3$ $s^{-1}$.

In case of OH(5-3)+OH(4-2) VER, presented in Fig. 5b, the new approach leads to a weak underestimation of the observations by the model in the altitude region above 85 km, even if $OH(v=9)+O(^3P)$ of R11 solely populates $OH(v=4)$. The model results are most sensitive to $k_{11}(v=5)$, and therefore this rate might be too fast. Considering our best fit value obtained for $k_{11}(v=6)$, it is indicated that $k_{11}(v)$ decreases with decreasing vibrational level and this feature is discussed below in more detail.

Thus, an upper limit of $k_{11}(v=5)<k_{11}(v=6)$ is recommended and the actual rate coefficient has to balance the direct contribution of $OH(v=9)$ to $OH(v=4)$ via R11. Investigating another scenario of $k_{11}(v=5)$ being zero showed that the branching of $OH(v=9)$ to $OH(v=4)$ has to be at least about 0.6 which corresponds to a rate of a $\sim1.4\times10^{-10}$ $cm^3$ $s^{-1}$.

The assumption that $k_{11}(v)$ decreases at lower vibrational levels is supported by the overall rate of

$OH(v=7)+O(^3P)\rightarrow OH(v')+O(^1D)$ at mesopause temperature which is suggested to be on the order of $0.9\text{-}1.6\times10^{-10}$ $cm^3$ $s^{-1}$ (Thiebaud et al., 2010; Varandas, 2004). At least to our knowledge, the total rate of $OH(v=8)+O(^3P)\rightarrow OH(v')+O(^1D)$ was not measured. Nevertheless, results reported by Mlynczak et al. (2018) and Panka et al. (2017, 2018) indicate that this rate might be slower than the value of $2.3\times10^{-10}$ $cm^3$ $s^{-1}$ suggested by Sharma et al. (2015). This is also in agreement with our findings here, because

applying $2.3\times10^{-10}$ $cm^3$ $s^{-1}$ for $k_{11}(v=9,8)$ results in non-physical $[O(^3P)]$ values above 90 km. The corresponding value of $[O(^3P)]$ e.g. at 95 km is about 1.25 times larger than SABER $[O(^3P)]$ 2013

(Mlynczak et al., 2013a) which in turn is about 1.15 times larger than the upper limit of [O($^3$P)] (Mlynczak et al., 2013b, their Fig. 4). This results in a factor of $1.15 \times 1.25 = 1.44$ (=44 %) above the upper limit and cannot be explained by the uncertainty of the [O($^3$P)] profile derived here (40 %, see Sect. 3.4). In order to obtain reasonable [O($^3$P)] values, it was necessary to lower the rate of $k_{11}(\nu=8)$ to $1.8 \times 10^{-10}$ cm$^3$ s$^{-1}$, and we therefore recommend $k_{11}(\nu=8) \leq 1.8 \times 10^{-10}$ cm$^3$ s$^{-1}$ as an upper limit to derive physically allowed [O($^3$P)] values.

It is seen in Fig. 5c that observations and O$_2$ best fit O($^3$P) $\nu$-5 model output of OH(3-1) VER are in agreement within the corresponding measurement errors but the model values seem to be slightly too low at heights >85 km. In this altitude region, simulated OH(3-1) VER is most influenced by OH($\nu=9,8$)+O($^3$P) of R11 because both vibrational levels can directly populate OH($\nu=3$). However, not much is known about the individual branching ratios of R11 except that OH($\nu=9$)+O($^3$P)→OH($\nu=3$)+O($^1$D) is an important deactivation channel but not necessarily the dominating one (Kalogerakis et al., 2016). These authors suggested a rate of $2.3(\pm 1.0) \times 10^{-10}$ cm$^3$ s$^{-1}$ and noted that this rate might be slower due to the involvement of excited surfaces. This generally agrees with our results presented here because the O$_2$ best fit O($^3$P) $\nu$-5 model only considers a contribution of OH($\nu=8$) to OH($\nu=3$) and the underestimation indicated in Fig. 5c could be attributed to the missing channel OH($\nu=9$)+O($^3$P)→OH($\nu=3$)+O($^1$D). The conclusions drawn from comparisons between three different airglow observations and our model studies with respect to OH($\nu$)+O($^3$P) quenching are summarized in Table 3.

Finally, all these findings presented in Table 2 and 3 were adapted in the "Best fit model" (Fig. 5, red lines), resulting in an overall agreement between model output and measurements within the corresponding errors. Note that $k_{11}(\nu=7)$ used here is the average of the lower and upper limits derived from Thiebaud et al. (2010) and Varandas (2004) which is unlikely to be seriously in error. Furthermore, we have to point out that lowering $k_{11}(\nu=8)$ does only impact the [O($^3$P)] and [H] derived here but does not affect the general conclusions drawn in this section.

The empirically determined solution presented here implies that the contribution of OH($\nu=9$) to OH($\nu=8$) via quenching with O($^3$P) is close to zero (see Table 1 and this section). In contrast, the model

described in Mlynczak et al. (2018) assumes single-quantum relaxation (OH($v$=9)+O($^3$P)→OH($v$=8)+O($^3$P)) to get the global annual energy budget into near balance. But applying this approach in our OH model (same total rate of $3\times10^{-10}$ cm$^3$ s$^{-1}$ and varying the rates for OH($v\leq8$)+O($^3$P)) leads to a considerable overestimation of OH(6-2) VER. Additionally, the shape of simulated OH(5-3)+OH(4-2) VER slightly mismatches the observed OH(5-3)+OH(4-2) VER above 90 km (not shown here). Based on these sensitivity runs, we conclude that at least part of OH($v$=9)+O($^3$P) channel has to be deactivated via multi-quantum quenching. This is supported by the results presented by Panka et al. (2017) which adjusted an OH airglow model to fit night-time CO$_2$($v_3$) emissions at 4.3 μm. However, this study reported empirically determined rates for OH(5$\leq v\leq$8)+O($^3$P) generally higher than the rates obtained in this work. But these differences might be attributed to their faster values of OH($v$)+O$_2$ because they seem to have falsely assumed that the rates of Adler-Golden (1997) do not take mesopause temperature into account. Thus, we think that their rates of OH($v$)+O$_2$ are too high, at least by a factor of ~1.5. Since they performed an empirical study, it is not possible to estimate how much this issue affects the rates of OH(5$\leq v\leq$8)+O($^3$P). But we know from our work that higher rates of OH($v$)+O$_2$ lead to higher values of OH(6-2) VER, OH(5-2)+OH(4-2) VER, and OH(3-1) VER which can be generally balanced by higher rates of OH(5$\leq v\leq$8)+O($^3$P). Considering our comparisons with these two studies, we think that the rates of OH($v$)+O($^3$P) should be investigated in more detail in future studies as this rate has a huge impact on derived values of [O($^3$P)] (Panka et al., 2018).

## 3.4 Derived profiles of [O($^3$P)] and [H]

Figure 6 displays the vertical profiles of [O($^3$P)] and [H] obtained by the Best fit model in comparison with the results derived from SABER OH(9-7)+OH(8-6) VER only (Mlynczak et al., 2018). The [O($^3$P)] profiles seen in Fig. 6a agree below 85 km but the Best fit model shows gradually larger values in the altitude region above. Between 85 km and 95 km, these larger values are caused by the different deactivation rates and schemes of OH($v$)+O($^3$P), agreeing with general pattern reported in Panka et al. (2018). We have to point out that other studies (e.g. von Savigny and Lednyts'kyy, 2013) observed a pronounced [O($^3$P)] maximum of about $8\times10^{11}$ cm$^{-3}$ at 95 km. The [O($^3$P)] derived here indeed shows

similar values at 95 km but a maximum is not seen. Nevertheless, the [O($^3$P)] in our study obtained above 95 km looks rather unexpected and possible reasons are discussed below.

The night-time [H] derived in this study shows similar pattern as SABER [H], including the maximum at 80 km. But Best fit model [H] is systematically larger than SABER [H] by a factor of approximately 1.5. This is primarily caused by our faster OH(v=8)+O$_2$ rate compared to the rate applied in Mlynczak et al. (2018). Similar to the comparisons with [O($^3$P)], Best fit model [H] results also shows unexpected patterns above 95 km.

The quality of the derived profiles is primarily affected by three different uncertainty sources. The first source includes uncertainties due to the rates of chemical and physical processes as well as the background atmosphere considered in the Best fit model. We assessed the 1$\sigma$ uncertainty by assuming uncorrelated input parameters. Adler-Golden (1997) did not state any uncertainties for $f_9$ and $f_8$ but these values should be similar to the uncertainty of $f_8$ derived by Klenerman and Smith (1987). Therefore, we applied an uncertainty of 0.03 for $f_9$ and $f_8$. In case of the Einstein coefficient, we adapted an uncertainty of 30 %, which is based on the five sets of Einstein coefficients analyzed in Sect. 3.1. Note that larger uncertainties only occur for multi quanta transitions >2 quanta. But [O($^3$P)] and [H] were calculated from the transition OH(9-7)+OH(8-6) where the agreement is better. All the other 1$\sigma$ uncertainties of the input parameters were taken from their respective studies.

Recent comparisons between MIPAS O$_3$ and SABER O$_3$ derived at 9.6 µm were performed by Lopez-Puertas et al. (2018). The authors showed that night-time O$_3$ from SABER is slightly larger than night-time O$_3$ obtained from MIPAS in the altitude region 80-100 km over the equator (their Fig. 8 and 10) but these differences are within the corresponding errors. Thus, at least to our knowledge there is no conclusive evidence stating that SABER night-time O$_3$ is generally too large. Nevertheless, we considered an uncertainty of O$_3$ of about 10 % (Smith et al., 2013). The uncertainty of SABER temperature was estimated to be lower than 3 % (Garcia-Comas et al., 2008) while the total uncertainty of SABER OH(9-7)+OH(8-6) VER was assumed to be about 6 % (see Sect. 2.1.2). The total 1$\sigma$ uncertainty was obtained by calculating the root-sum-square of all individual uncertainties. The results of 1$\sigma$ uncertainty of [O($^3$P)] and [H] derived by the Best fit model are shown as error bars in Fig. 6. The

error bars of SABER [$O(^3P)$] and [H] were adapted from the corresponding publication.

In case of the Best fit model [$O(^3P)$] profile, the $1\sigma$ uncertainty varies between 30 % and 40 %, depending on altitude. The individual contributions of the input parameters to the total $1\sigma$ uncertainty are considerably different. Einstein coefficients and nascent distribution each account for about 10 % and 5 %, respectively, throughout the entire height interval. The influence of the collision rates is about 5 % and gradually decreases to zero with increasing altitude. In contrast, the chemical reaction rates $k_2$ and $k_3$ account for ~80 % to ~85 % of the overall $1\sigma$ uncertainty of the derived [$O(^3P)$] profiles. The total $1\sigma$ uncertainty of [H] varies between 25 % and 40 % with $k_1$ being the major uncertainty source (~80 %) below 85 km. In higher altitude regions, the impact due to uncertainty of [$O(^3P)$] becomes gradually more important and both $k_1$ and [$O(^3P)$] each contribute close to one half to the overall uncertainty at altitudes >95 km. We further assumed a worst case scenario (not shown here), meaning that all uncertainties of the input parameters contribute to either higher or lower [$O(^3P)$] values, obtaining a worst case $1\sigma$ uncertainty of approximately 80 % for [$O(^3P)$] and about 65 % for [H]. However, it is more likely that the uncertainties are uncorrelated since they originate from independent measurements.

The second aspect influencing the quality of the derived profiles is the assumption of chemical equilibrium of $O_3$, represented by Eq. (3). This issue was recently investigated by Kulikov et al. (2018), which carried out simulations with a 3-D chemical transport model and demonstrated that a wrongly assumed chemical equilibrium of $O_3$ may lead to considerable errors of derived [$O(^3P)$] and [H]. In order to test the validity of chemical equilibrium of $O_3$ locally, the authors suggested that OH(9-7)+OH(8-6) VER has to exceed $10 \times G \times B$, with $B$ including several chemical reaction rates involving $O_x$ and $HO_x$ species. Note that this criterion requires simultaneously performed temperature and OH airglow measurements. Furthermore, this criterion is based on the assumption that the impact of atmospheric transport on chemical equilibrium of $O_3$ is negligible. Since our experiments fit these conditions, we applied their suggested limit and found that in our case chemical equilibrium of $O_3$ is valid above 80 km. We have to point out that the term "chemical equilibrium of $O_3$" refers to $O_3$ that does not deviate more than 10 % from $O_3$ in chemical equilibrium (Kulikov et al., 2018, their Eq. 2). Assuming that $O_3$ is always 10 % greater or lesser than $O_3$ in chemical equilibrium introduces an

uncertainty of about 10 % at 80 km and 20 % at 95 km, additionally to the total uncertainty of [O($^3$P)] and [H] estimated above. However, such a worst case scenario is rather unlikely while it is more realistic that O$_3$ actually varies around its chemical equilibrium concentration. Thus, an over- and underestimation of derived [O($^3$P)] and [H] are assumed to compensate each other. Consequently, we conclude that the impact on the total uncertainty of [O($^3$P)] and [H] due to deviations from chemical

equilibrium of O$_3$ is negligible, but only because the previously used criterion (OH(9-7)+OH(8-6) VER>10×G×B) is valid.

The last problem lies in the fact that the approach used here (see Sect. 2.2) has to be applied to individual OH airglow profiles to derive [O($^3$P)] and [H] correctly. However, the individual scans of OH(6-2) were too noisy to analyze single profiles and we therefore used climatology for all input

parameters. By investigating individual OH airglow profiles, we would derive individual [O($^3$P)] profiles and eventually average them to the mean [O($^3$P)] profile. While in our case, we directly derive the mean [O($^3$P)] profile. This makes no difference as long as the relation between OH airglow and [O($^3$P)] is a linear one. But Eq. (4b) shows that the relation between [O($^3$P)] and OH(9-7)+OH(8-6) VER is only approximately linear because $G$ also depends on [O($^3$P)], as represented by the terms $C_v$

and $C_{vv}$. The linearity between OH(9-7)+OH(8-6) VER and [O($^3$P)] of an air parcel with a certain temperature and pressure is solely controlled by [O($^3$P)]×$G$. Note that [H] too is affected by this non-linearity issue since [H] depends on $G$ (Eq. (4a)). Thus, derived [H] values are only reliable as long as the derived [O($^3$P)], and as a consequence $G$, is not seriously in error.

In order to test the linearity, [O($^3$P)]×$G$ was plotted as a function of [O($^3$P)] and the corresponding

results for Best fit model at five different heights are presented in Fig. 7. It is seen that the relation between [O($^3$P)] and [O($^3$P)]×$G$ or OH(9-7)+OH(8-6) VER, respectively, is linear for small values of [O($^3$P)], while a non-linear behaviour becomes more pronounced for larger values of [O($^3$P)]. Furthermore, the starting point of the behaviour is shifted to lower [O($^3$P)] values at higher altitudes. In order to estimate this threshold, we performed a visual analysis and determined an upper limit of

[O($^3$P)] before non-linearity of [O($^3$P)]×$G$ takes over. The approximated upper limits are added as dashed lines in Fig. 7. Finally, an [O($^3$P)] value at a certain altitude is assumed to be true if this value is below the corresponding upper limit of [O($^3$P)]. Otherwise, it should be viewed more critically. This

was done for each altitude and we found that the $[O(^3P)]$ and $[H]$ profiles presented in Fig. 6 are plausible in the altitude region <95 km. In combination with the estimation of chemical equilibrium of

$O_3$ and the maximum of physically allowed $[O(^3P)]$, we think that the $[O(^3P)]$ and $[H]$ derived by the Best fit model are reasonable results between 80 km and 95 km. Note that these altitude limits do not affect the results with respect to $OH(v)+O_2$ and $OH(v)+O(^3P)$ presented in the Sect. 3.2 and 3.3.

**4 Conclusions**

We presented a zero dimensional box model which fits the VER of four different OH airglow

observations, namely TIMED/SABER OH(9-7)+OH(8-6) and OH(5-3)+OH(4-2) as well as ENVISAT/SCIAMACHY OH(6-2) and OH(3-1). Based on a night-time mean zonal mean climatology of co-location measurements between 2003 and 2011 at 0°-10° N, we found that I) $OH(v)+O_2$ is likely to occur via multi-quantum deactivation while $OH(v\geq7)$ primarily contribute to $OH(v\leq5)$ and might prefer deactivation to $OH(v'=v-5)+O_2$. This relaxation scheme generally agrees with results reported in

Russell et al. (2005) but is considerably different to the commonly used scheme suggested by Adler-Golden (1997). We further found II) general support for the new pathway $OH(v)+O(^3P)\rightarrow OH(v')+O(^1D)$ proposed by Sharma et al. (2015) but suggest slower total loss rates of $OH(v=8,7,6,5)+O(^3P)$. Additionally, hints for a favoured deactivation to $OH(v'=v-5)+O(^1D)$ are obtained.

We have to stress that we performed an empirical model study and the total rates and deactivation channels suggested here heavily depend on the OH transitions considered. Including additional OH transitions, like OH(9-4), OH(8-3), and OH(5-1) from the Optical Spectrograph and InfraRed Imager System (OSIRIS) on board the Odin satellite, might result in other values and deactivation schemes. This could be a subject of a future study. Also note that the Einstein coefficients used here might be in

error (see Sect.3.1; Fig. 2). This does not affect the two general conclusions drawn above but would impact the empirically derived rates.

Furthermore, our OH airglow model is based on the transitions OH(9-7)+OH(8-6), OH(6-2), OH(5-3)+OH(4-2), and OH(3-1) only. Therefore, our model does not provide any information of $OH(v\leq2)$. It

further cannot distinguish between OH(v=5) and OH(v=4) as well as OH(v=9) and OH(v=8), respectively, and errors in OH(v=5) and OH(v=9) might be compensated by errors in OH(v=4) and OH(v=8) or vice versa. Consequently, the rates of the individual deactivation channels presented in Table 2 and Table 3 should be viewed as a suggestion only. In particular, the rate of OH(v=9)+O($^3$P)→OH(v=3)+O($^1$D) is about 3 times slower than the lower limit reported by Kalogerakis et al. (2016). But these issues will only be solved eventually when future laboratory experiments provide the corresponding OH(v)+O$_2$ and OH(v)+O($^3$P) relaxation rates and deactivation channels. Nevertheless, we have to emphasize that the shortcomings of our model do not affect the two main conclusions drawn in this study.

Justified by a nearly linear relation between [O($^3$P)] and OH(9-7)+OH(8-6) VER, the physically allowed upper limit of [O($^3$P)], and also considering the chemical equilibrium of O$_3$, we conclude that the [O($^3$P)] and [H] profiles derived by the Best fit model are plausible in the altitude range from 80 km to 95 km. The corresponding 1σ uncertainty due to uncertainties of chemical reactions and physical processes varies between 35 % and 40 % ([H]) and between 30 % and 40 % ([O($^3$P)]), depending on altitude.

The [H] derived here is systematically larger by a factor of 1.5 than SABER [H] reported in Mlynczak et al. (2018) which is primarily attributed to their slower OH(v=8)+O$_2$ rate. Our [O($^3$P)] values in the altitude region below ~87 km are in agreement within the corresponding errors with the results found in Mlynczak et al. (2018) and Zhu and Kaufmann (2018) but are lower than the values presented in Panka et al. (2018). However, we think that the results of the latter study are too large because the authors falsely assumed too fast OH(v)+O$_2$ rates. In the altitude region above ~87 km, the [O($^3$P)] shown here is generally larger than the values reported in these three studies up to a factor 1.5 to 1.7. These differences are attributed to the faster rates and different deactivation channels of OH(v)+O($^3$P). Therefore, it is indicated that we might overestimate [O($^3$P)] above >87km and we suggest that our results should be interpreted as an upper limit. However, a final conclusion cannot be drawn at this point due the large uncertainties of the rates assumed to derive [O($^3$P)].

## Data availability

The data used in this study are open for public. The TIMED/SABER data can be downloaded from http://saber.gats-inc.com/data.php while ENVISAT/SCIAMACHY can be accessed by getting in contact with Christian von Savigny (csavigny@physik.uni-greifswald.de).

## Author contribution

MS initialized and supervised the study. CVS retrieved the SCIAMACHY data. TF performed the model runs and wrote the final script with contributions from all co-authors.

## Competing interests

The authors declare that they have no conflict of interest.

## Acknowledgments

T. Fytterer and M. Sinnhuber gratefully acknowledge funding by the Deutsche Forschungsgemeinschaft (DFG), grant SI 1088/6-1. The authors also acknowledge support by the Open Access Publishing Fund of Karlsruhe Institute of Technology.

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

**Table 1**. Physical processes and chemical reactions included in the Base model

| Process | | | | | | Rate or scheme | Reference |
|---|---|---|---|---|---|---|---|
| R1 | H | $+ O_3$ | | $\rightarrow OH(\nu)$ | $+ O_2$ | $k_1 = 1.4\times10^{-10}\ e^{(-470/T)}$ $k_1(\nu) = k_1\ f_1(\nu)^a$ | Burkholder et al. (2015), Alder-Golden (1997, Table 1) |
| R2 | $O(^3P) + O_3$ | | | $\rightarrow O_2$ | $+ O_2$ | $k_2 = 8\times10^{-12}\ e^{(-2060/T)}$ | Burkholder et al. (2015) |
| R3 | $O(^3P) + O_2 + M$ | | | $\rightarrow O_3$ | $+ M$ | $k_3 = 6\times10^{-34}\ (300/T)^{2.4}$ | Burkholder et al. (2015) |
| R4 | $OH(\nu)$ | | | $\rightarrow OH(\nu') + h\nu$ | | variable rates | Xu et al. (2012, Table A1) |
| R5 | $OH(\nu)$ | $+ N_2$ | | $\rightarrow OH(\nu') + N_2$ | | $\nu'=\nu-1$ | Adler-Golden (1997, Table 1), Kalogerakis et al. (2011) |
| R6 | $OH(\nu)$ | $+ O_2$ | | $\rightarrow OH(\nu') + O_2$ | | $\nu'<\nu$ | Adler-Golden (1997, Table 3), see text for more information |
| R7a | $OH(\nu)$ | $+ O(^3P)$ | $\rightarrow$ | H | $+ O_2$ | variable rates | Varandas (2004, Table 3, M I) |
| R7b | $OH(\nu)$ | $+ O(^3P)$ | $\rightarrow$ | $OH(\nu') + O(^3P)$ | | $\nu'<\nu$ | Caridade et al. (2013, Table 1) |

Rate constants are given in the $cm^3$-$s^{-1}$ system.
[a]$f_1(5, 6, 7, 8, 9) = 0.01, 0.03, 0.15, 0.34, 0.47$

**Table 2.** Empirically determined branching ratios of OH($\nu$)+$O_2$→OH($\nu$')+$O_2$ of the $O_2$ best fit model based on OH(6-2) VER, OH(5-3)+OH(4-2) VER, and OH(3-1) VER observations.

| $\nu/\nu'$ | 8 | 7 | 6 | 5 | 4 | 3 | ≤2 |
|---|---|---|---|---|---|---|---|
| 9 | 0 | 0 | 0 | 0 | 1 | 0 | 0 |
| 8 | | 0 | 0 | 0 | 0.3 | 0.7 | 0 |
| 7 | | | 0 | 0 | 0 | 0.1 | 0.9 |
| 6 | | | | 0 | 0 | 0 | 1 |
| 5 | | | | | 0 | 0 | 1 |
| 4 | | | | | | 0 | 1 |
| 3 | | | | | | | 1 |






**Table 3.** Empirically determined branching ratios of OH($\nu$)+O($^3$P)$\rightarrow$OH($\nu$')+O($^1$D) of the Best fit model based on OH(6-2) VER, OH(5-3)+OH(4-2) VER, and OH(3-1) VER observations.

| Process | Recommendation | Best fit rate (cm$^3$ s$^{-1}$) |
|---|---|---|
| R11a   OH(9) + O($^3$P) $\rightarrow$ OH(4)   + O($^1$D) | $k_{11}$(9-4) > 0.6$\times k_{11}(\nu$=9) | 0.8$\times$2.30$\times$10$^{-10}$ |
| R11b   OH(9) + O($^3$P) $\rightarrow$ OH(3)   + O($^1$D) | not negligible | 0.2$\times$2.30$\times$10$^{-10}$ |
| R11c   OH(8) + O($^3$P) $\rightarrow$ OH(3)   + O($^1$D) | $k_{11}(\nu$=8) < $k_{11}(\nu$=9) | 1.0$\times$1.80$\times$10$^{-10}$ |
| R11d   OH(7) + O($^3$P) $\rightarrow$ OH($\leq$2) + O($^1$D) | $k_{11}$(7-$\leq$2) < $k_{11}(\nu$=8) | 1.25$\times$10$^{-10}$ |
| R11fe   OH(6) + O($^3$P) $\rightarrow$ OH($\leq$1) + O($^1$D) | $k_{11}$(6-$\leq$1) < $k_{11}(\nu$=7) | 0.80$\times$10$^{-10}$ |
| R11gf   OH(5) + O($^3$P) $\rightarrow$ OH        + O($^1$D) | $k_{11}(\nu$=5) < $k_{11}(\nu$=6) | 0.40$\times$10$^{-10}$ |




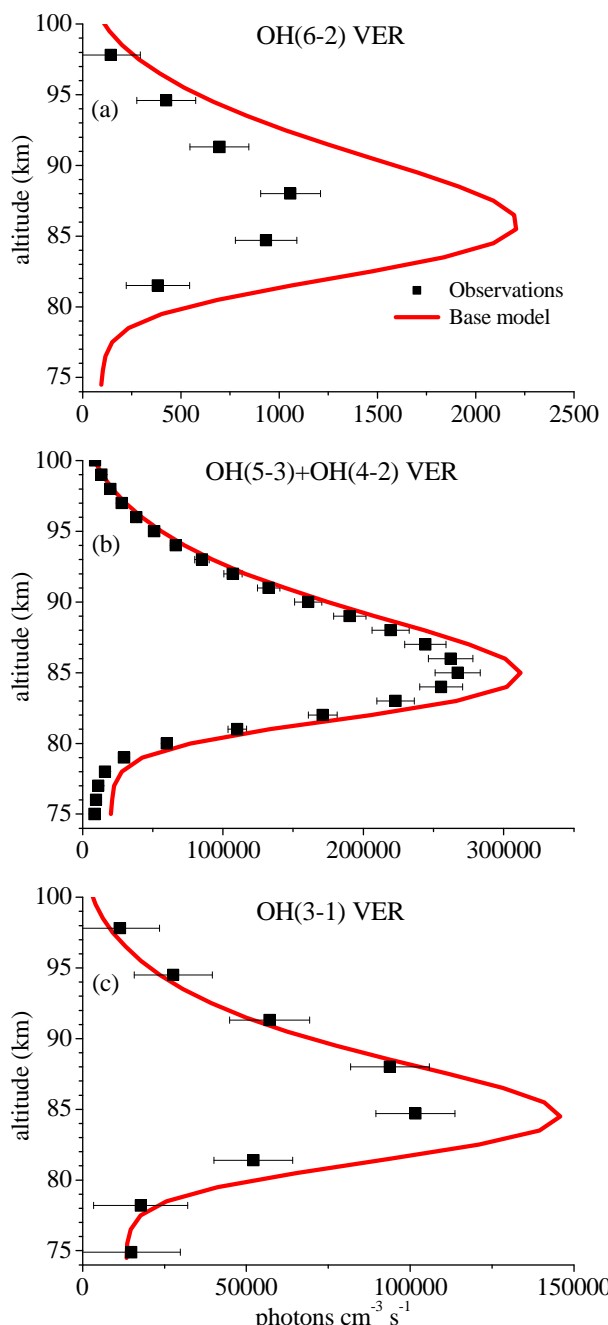

**Figure 1 : Comparison of vertical profiles of the volume emission rate (VER) of a) OH(6-2),  b) OH(5-3)+OH(4-2), and c) OH(3-1) at 0°-10° N between satellite observations and the Base model output. The observations are climatology of night-time mean zonal means from 2003 to 2011, based on co-location measurements of TIMED/SABER and ENVISAT/SCIAMACHY. Note the different scaling of the x-axis.**

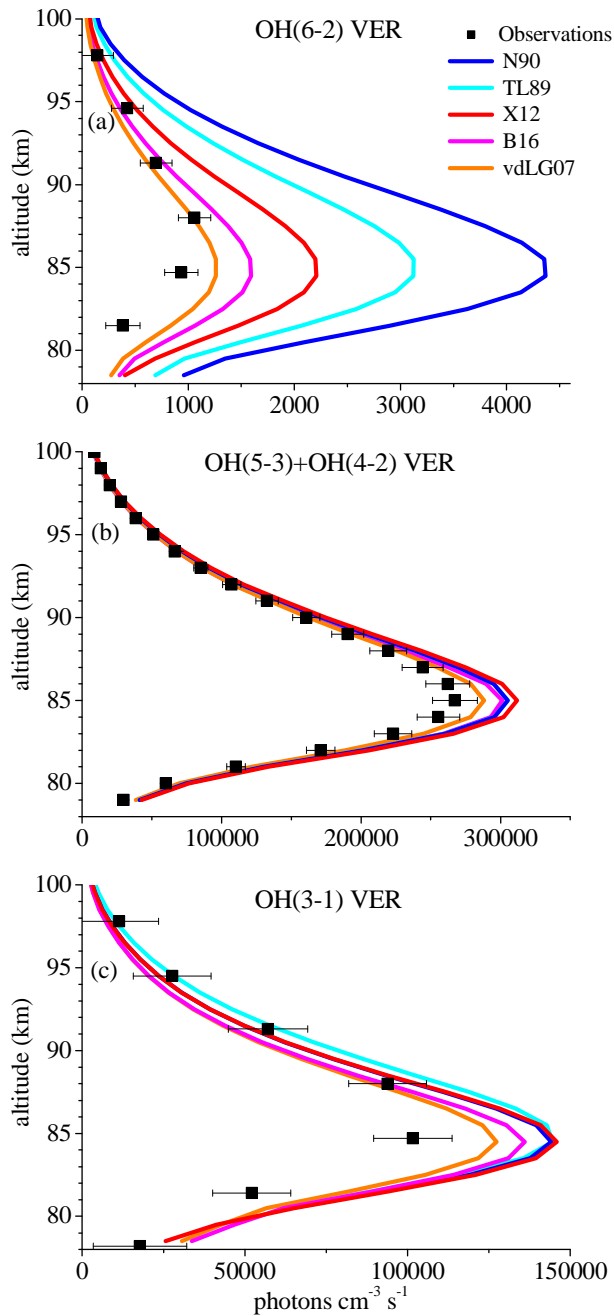

**Figure 2 : Same as Figure 1 but for different sets of Einstein coefficients from literature, namely N90 (Nelson et al., 1990), TL89 (Turnbull and Lowe, 1989), X12 (=Base model; Xu et al., 2012), B16 (Brooke et al., 2016), and vdLG07 (van der Loo and Groenenboom, 2007).**


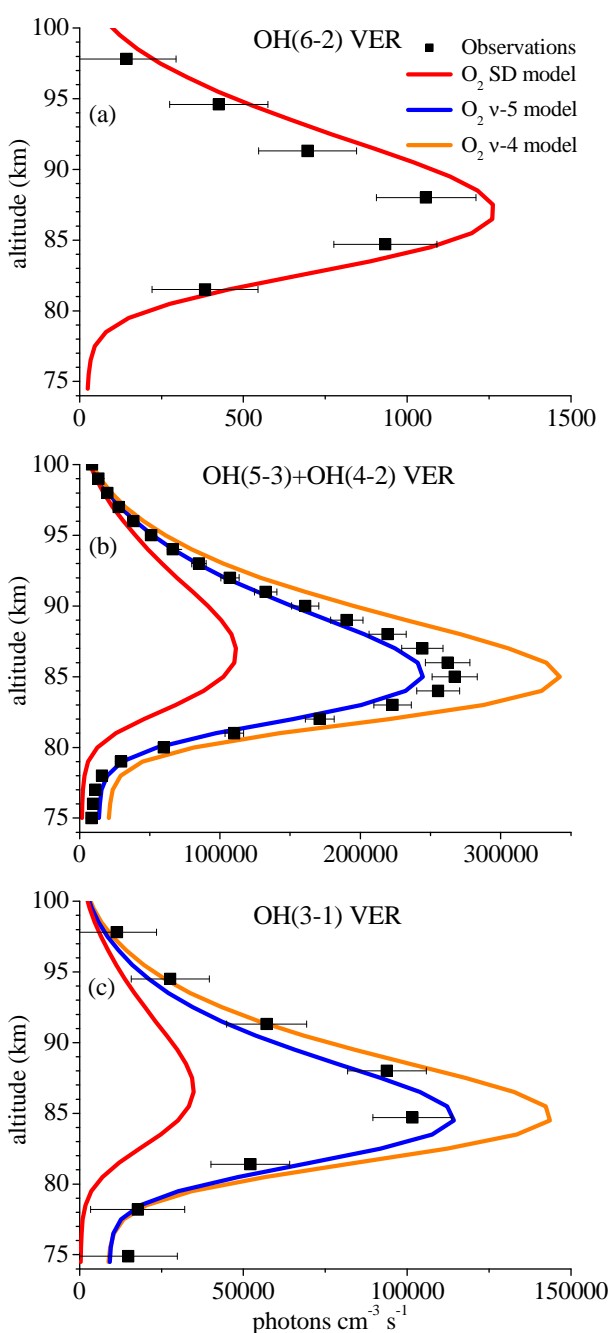

**Figure 3 : Same as Fig. 1 but for the O$_2$ SD model, the O$_2$ ν-5 model, and the O$_2$ ν-4 model. Note that the results of these three models are identical in case of OH(6-2) VER.**


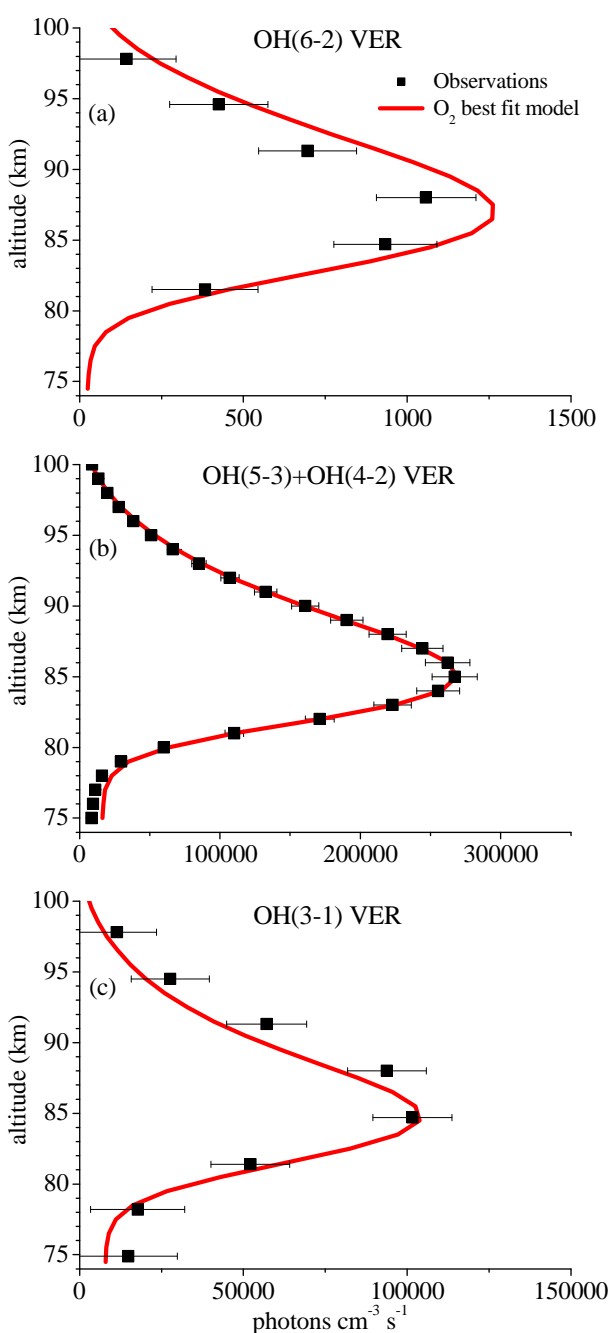

**Figure 4 : Same as Fig. 1 but for the O₂ best fit model. Note that Fig. 4a is identical to Fig. 3a but was plotted again for convenience.**


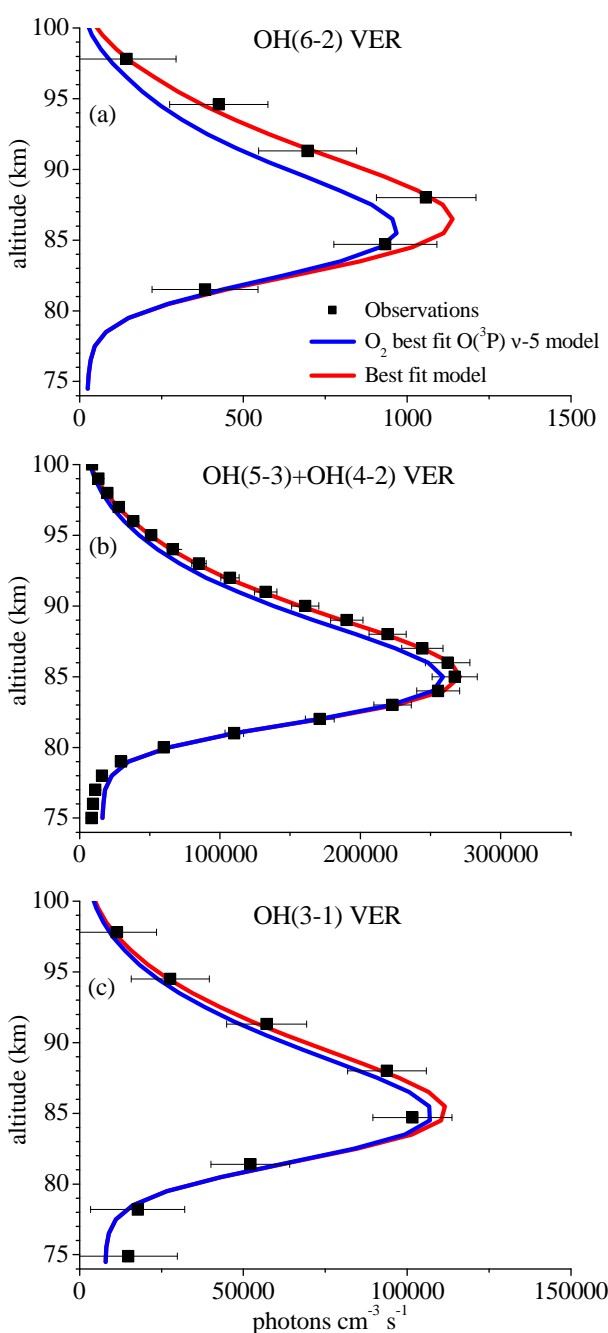

**Figure 5 : Same as Fig. 1 but for the O$_2$ best fit O($^3$P) v-5 model and the Best fit model.**


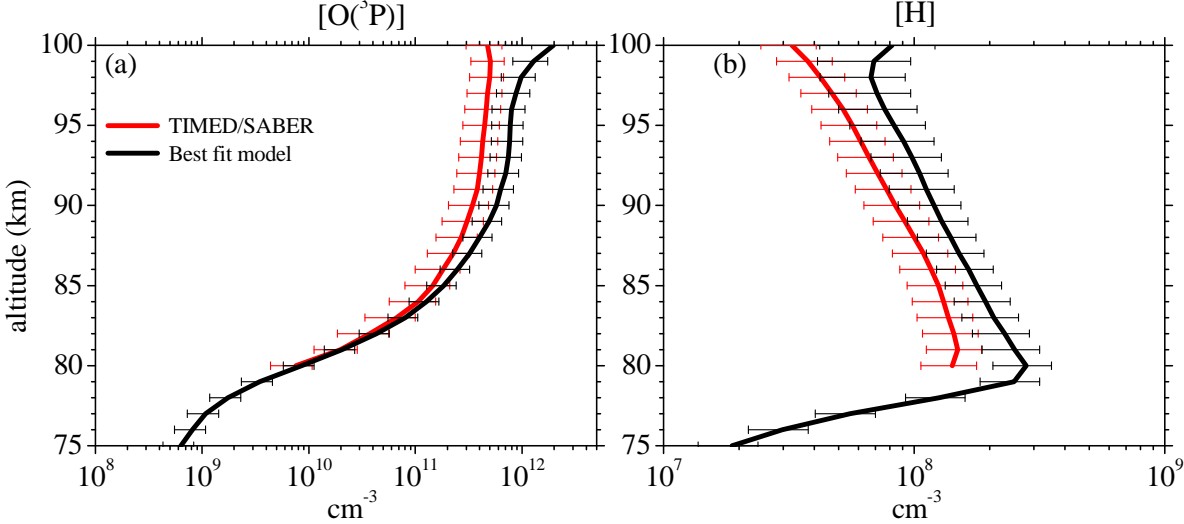

**Figure 6 :** Vertical profiles of a) [O($^3$P)] and b) [H] derived from SABER OH(9-7)+OH(8-6) VER observations (Mlynczak et al., 2018) and our Best fit model by fitting SABER OH(9-7)+OH(8-6) VER and OH(5-3)+OH(4-2) VER as well as SCIAMACHY OH(6-2) VER and OH(3-1) VER. Shown are averages of night-time mean zonal means of co-location measurements (see Sect. 2.2) from 2003 to 2011 between 0° and 10° N. Error bars show the 1σ uncertainty due to chemical and physical processes.

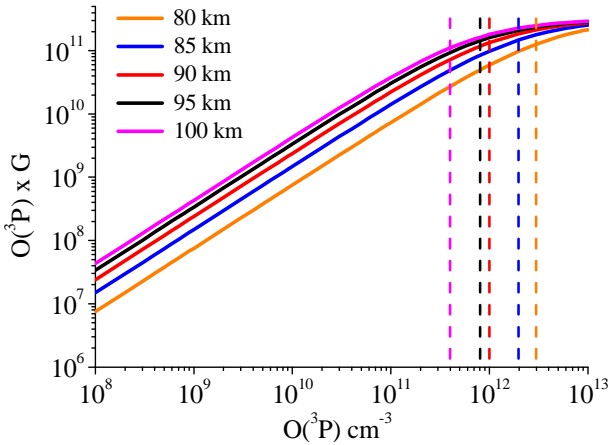

**Figure 7 : O($^3$P)×G as a function of O($^3$P) at different altitudes. The visually determined upper limits of O($^3$P) before non-linearity becomes too pronounced are represented by the dashed lines.**