# Peer review of "Model results of OH airglow considering four different wavelength regions to derive night-time atomic oxygen and atomic hydrogen in the mesopause region"

_Atmospheric Chemistry and Physics, 2018_

## Referee Comment (RC1) · Anonymous Referee #1 · 31 Aug 2018

This paper is another piece in the puzzle of OH(v>0) deactivation. It uses satellite data from SABER and SCIAMACHY in conjunction with a chemical box model to empirically determine optimal branching ratios and reaction rates for the reactions OH(v>0) + O2 → OH(v'<v) + O2 and OH(v>0) + O(3P) → OH(v'<v) + O(1D). The model is then used to show that it can be used to derive reasonable profiles of [O] and [H] in the mesopause region. It is a well written paper and the results are discussed in a very clear and methodical way, which makes it easy to follow the steps taken. As the results tend to strike a balance between many previous studies that have often been in opposition with each other, it seems like this would be a good fit for publication in ACP. I would recommend publication only after a few, relatively minor, issues have been properly

addressed, listed below.

General comments:

I know it's a bit persnickety, but throughout the paper you need to be careful distinguishing between X and [X], as is done in the equations. X is not being derived, you are deriving X densities, or deriving [X].

If O3 is also a variable in the airglow model, could you not compare the resulting O3 with SABER values as a further constraint, in addition to the SABER VER? Either way, it would be interesting to see how the best fit model O3 compares to the SABER values, since those are not related to OH (although if it is expected that SABER O3 values are too large, maybe this wouldn't work. Or could you compare to SABER O3 1.27 $\mu$ data?).

Also, please comment on how initial conditions of the target species affect the results of the model, i.e. have you tested this, what are the scale of any uncertainties the first guesses can add?

Specific comments:

Abstract should specifically indicate that the [O] and [H] profiles derived in this study are from the SABER observations using an OH model informed by SCIAMACHY and SABER observations.

L24: "high" should be "large" (as to not confuse with altitude)

L45 and onward: What is meant by "OH(v)"? Do you mean vibrationally excited OH? It should be defined when it is first used as "vibrationally excited OH" or "OH(v>0)".

L58: "last decades" sounds ominous. Should be specific, i.e. last three to four decades.

L64: "of" should be "from"

L74: "individually" doesn't sound right. Maybe, "Both airglow emissions were used to

derive separate data sets of O(3P) profiles"?

L75: "profiles" makes it sound as if only one profile was retrieved for each airglow feature. Should probably be "data sets".

L84: should be OH(v=9). Or define that OH(x) means OH(v=x).

L117: please fix the significant digit mismatch for "837.5-848"

L153: by "issues" do you mean uncertainties?

L176-177: should specify that the three-body reaction is the production of O3.

L179: M is not the total density of air, M represents an air molecule. [M] is the total density.

L184-185: The wording makes it sound as if the SABER O3 was derived via the ideal gas law. Did you mean to say that you're using SABER derived O3?

L202-203: would suggest "well suited" as opposed to "very suited".

L216-217: the way this sentence is worded means that the equation should be v=9. If that's not the case, it should read something like "OH at all vibrational levels $v \leq 9$"

L232: I assume that by "added" you mean "applied" and not literally added.

Figures 1-4: Why are there no error bars on the SABER observations?

L338: "probably" is not needed

L343-344: They also seem to match within the error bars above $\sim$92 km.

L344-345: I believe this sentence is missing an altitude value and a very important comma. Are you intending to say, "The model still overestimates the measurements in the altitude region above xx km, which might be related to O(3P) quenching."?

R8: this claims that you're only considering $0 \leq v' \leq v-5$, for $v \geq 6$. If that were the case, then the branching ratios for 8-4, and 7-3 should be 0, which, according to Table

2, they are not. Should it be $0 \leq v' \leq v\text{-}4$?

L374: "Including R10b in the model..." is confusing. In the v-4 scenario, are you including R10a and R10b, or are you including only R10b and not R10a. If it's the former, that would seem to imply that v'=v-4 can't occur at all (for $v \geq 6$), and then, again, the branching ratios for 8-4, and 7-3 should be 0. If it's the latter, then I agree that the implication is that v'=v-5 (and not v'=v-4) is the predominant pathway, which fits with the values in Table 2. Please make the explanation of this case clearer. (It's even more confusing in the context of R8, which already says this pathway isn't being considered.)

L402: Should be "that implied" instead of "which implied". Also, "implied" is somewhat vague and makes it sound like you might not be sure (same with "seems reasonable").

Table 3: Reactions 11a-d seem to indicate that k11 doesn't entirely decrease with v, which goes against what's written in the text. This is touched on a bit later, but not explicitly stated.

L459: "higher" should be "larger" as not to be confused with the discussion of altitude.

Figure 5: These plots would be much easier to read with boxed axes (ticks on the top and right). Also, this would be a good spot to compare O3 and show that the model O3 is (presumably) lesser than SABER values.

L540-541: Have you considered doing a similar study incorporating OH(9-4), (8-3), and (5-1) band VERs from OSIRIS?

Summary: needs a bit more description at the end of how [O] and [H] compare to the SABER results and explaining the differences.

---

## Referee Comment (RC2) · Anonymous Referee #2 · 3 Sep 2018

General comments:

This study proposes a new OH airglow model to retrieve O and H densities in the mesosphere. The OH model is empirically developed to simultaneously fit four OH emissions observed by the SABER/TIMED instrument at 2.0- and 1.6-microns as well as the OH(6-2) and OH(3-1) bands measured by SCIAMACHY/ENVISAT. The authors show that using adjusted rate coefficients and specific state-to-state relaxation mechanisms, the OH model reproduces the four emissions. However, they retrieve very high O and H concentrations.

The concept of fitting four emission bands simultaneously is promising as it may con-

strain unknown parameters involved in modeling OH emissions. The conclusions regarding their new OH model, however, are speculative as model inputs used in the study to simulate emissions have larger uncertainties than the authors claim (i.e. Einstein coefficients, ozone concentration). Accounting for these uncertainties will significantly alter the results of this paper. Further, the authors show that the applied OH model retrieves unrealistically high atomic oxygen and hydrogen in the MLT. Recent publications (s.f. Kaufmann et al. 2014, Mlynczak et al. 2018, Panka et al. 2018, and Zhu and Kaufmann 2018) have shown that [O] densities retrieved using SABER and SCIAMACHY measurements are much lower (by up to a factor 2 and more) than those retrieved in this study, specifically from 85-100 km.

The model development (and the rate coefficient adjustments) must have a goal to reliably retrieve atmospheric properties from the observations. The very high [O] and [H] retrieved with the help of the new model indicates that there are still major flaws (it does not matter that it fits all selected emissions, this system has a very large number of unconstrained variables). The paper needs major revisions before making physical sense and being suitable for publication.

Instead of fitting four emission bands while simultaneously retrieving [O] and [H] densities, I recommend for the revised study, to concentrate on retrieving [O] and [H] densities but only fit three emission bands (as will be discussed below, the OH(6-2) emission band is unreliable and taking into account its large uncertainty will alter the results of the current study). Further, the authors must demonstrate how the rates derived from zonal mean profiles fit real single scans in three emission bands.

Specific comments:

Line 1. The title states "New insights in OH airglow modeling...". The proposed new "insights" are highly speculative and are inconsistent with existing theory and experiments. The authors need to first show that reliable [O] and [H] can be derived when their OH model is applied before claiming any new insights.

In the introduction section, discussion regarding the current progress of [O] and [H] retrievals using SABER and SCIAMACHY instruments is missing. Retrieving these two parameters are a key point of this study and no background is given. Please cite recent [O] and [H] retrieval studies and their proper discussion.

Lines 175-185. The retrieval of [O] and [H] are both dependent on the [$O_3$] volume mixing ratio. The authors used nighttime [$O_3$] taken from SABER. The nighttime SABER [$O_3$] has never been rigorously validated, nor is there a paper discussing its retrieval approach. Differences between WACCM and SABER [$O_3$] are roughly a factor of 2 (Smith et al. 2014) and, therefore, one cannot rely on SABER [$O_3$] as an input parameter. Additionally, in Mlynczak et al. 2018, the conclusion is made that current SABER daytime [$O_3$] and, supposedly, the nighttime one is too high based on a significantly lower [O] retrieved in that study. It is clear from equation 4a and 4b that any variation in [$O_3$] will have a significant effect on [O] and [H]. For the revised study, I recommend using inputs taken from a self-consistent photochemical model like WACCM instead of ones taken from retrievals, which are not supported by any other studies. Additionally, uncertainties in the retrieved parameters due to large uncertainties in the [$O_3$] must be estimated and discussed.

Lines 268-269. "...we exclude the Einstein coefficients as a potential fundamental error source." I do not agree with this statement. The new constraint imposed using the OH(6-2) emission band is unreliable. This band has a very small Einstein coefficient. The authors do not go into detail regarding the numerical differences among the literature of the OH(6-2) emission rate, but state that they are consistent. The authors use the OH(6-2) coefficient taken from Xu et al. 2012 which is 1.767 sec$^{-1}$. A more recent publication by Brooke et al. (2015) recalculated OH Einstein coefficients and found a rate of 1.16 sec$^{-1}$ for the same transition. The rate of Xu et al. (2012) is approximately 50% larger than that of Brooke et al. (2015) and would significantly change the OH(6-2) emission profiles in Figures 1-4 as well as the results in Tables 2 and 3. The ab initio calculations of van der Loo and Groenenboom (2007, 2008) give values

that are even smaller than Brooke et al. (2015) - the OH(6-2) emission rate of Xu et al. (2012) is 75% larger than that of van der Loo and Groenenboom (2007, 2008). Evidently, the issue of the OH Einstein coefficients is not yet settled. To make matters worse, the SCIAMACHY OH(6-2) band displays a signal count two orders of magnitude smaller than that of the OH(3-1) band. The uncertainty of the VER signals for these two bands will be vastly different. Finally, because these two bands have a different Delta v and values of Einstein coefficients that are different by more than one order of magnitude, the uncertainty of the OH(6-2) band will be much larger. Therefore, both the observed low VER signals and the large uncertainty in the value of the Einstein coefficient indicate knowledge of the OH(6-2) band is highly uncertain. As a result, the OH(6-2) emission band cannot be confidently used to constrain OH modeling parameters. For manuscript revisions, I recommend to redo this study using only three OH emission bands. An alternative would be to validate the intensity of the OH(6-2) band by comparison with the OH(6-3) profile, which should be within the capabilities of SCIAMACHY.

Lines 493-498. "Applying their suggested limit, we found that in our case chemical equilibrium of O3 is probably true only above 80 km." Recent studies have shown that the [O] and [H] retrieval approach used in this study may be flawed (Belikovich et al., 2018; Kulikov et al., 2017, 2018) and can introduce additional uncertainties. The authors addressed these issues very briefly here, but this needed to be more rigorously discussed. To say just simply "probably true" is insufficient. Additionally, uncertainties of the final results related to a probable chemical equilibrium breakdown need to be estimated and discussed.

Lines 521-522. "...we think that the O($^3$P) and H derived by the Best-fit model provides reasonable results between 80 and 95 km." The [O] derived looks somewhat reasonable only below 87 km, but not above this altitude. At 95 km, the retrieved [O] is at least two times larger than Mlynczak et al. [2018] and more than a factor of 5 at 100 km. It has also been discussed in detail that high [O] will disrupt the energy

balance in the MLT (Mlynczak et al. 2013, 2018) and influence temperature retrievals. If, in the revised study, the retrieved [O] and [H] remain high, then please demonstrate how it impacts the heating and cooling of MLT and discuss in detail possible ways to overcome the corresponding energy budget imbalance.

Line 537-539. "Furthermore, it cannot distinguish between OH(5) and OH(4) as a well as OH(9) and OH(8), and consequentially errors in OH(5) and OH(9) might be compensated by errors in OH(4) and OH(8) or vice versa". This is a troubling statement as your main results in Table 3 (R11a, R11b, R11c, R11d, and R11g) involve these levels and describe rate coefficients for specific state-to-state reactions. This statement needs to be clarified. It sounds as if you treat OH(9)+OH(8) as a combined, single level as well as OH(5)+OH(4). Is this true? If you cannot distinguish between certain vibrational levels, then how can you determine rate coefficients for specific vibrational levels?

Tables 2  3. It is not clear if the results in Table 2 and 3 describe the Best-Fit model discussed in the conclusion. Table 2 shows empirically determined branching ratios of the OH(v) + O$_2$ reaction for only VER observations "below 85 km" while Table 3 shows the branching ratios of the OH(v) + O($^3$P) reaction for only VER observations "above 85 km". The lack of consistency adds confusion to the findings of this study. Please clarify this. Is there not a best-fit model for altitudes 80-100 km?

Table 3. The two most important processes (largest rate coefficients) estimated from the best fits are not energetically allowed! Processes R11a and R11c are highly endothermic processes by ∼3000 cm$^{-1}$ and 2000 cm$^{-1}$, respectively. Additionally, the state-to-state rate coefficients in Table 3 for the OH+O($^3$P) reaction appear to be in contradiction with the findings of Kalogerakis et al. (2016), who measured a large rate coefficient attributed to the resonant reaction OH(9)+O($^3$P) → OH(3)+O($^1$D). These results are non-physical and must be revised. As stated above, it seems most likely that fitting the highly uncertain OH(6-2) signal that has large systematic errors have skewed the results of this paper. Removing this constraint may bring the revised OH

[Figure]

model into better agreement with recent laboratory and modeling studies as well as retrieve reasonable [O] and [H].

Figures 1-4. Why are there no error bars displayed for the measured OH(5-3)+OH(4-2) VER emissions despite error bars displayed for the OH(6-2) and OH(3-1) emissions? In general, the concept of fitting the zonal mean profiles for three OH bands is questionable. Operating with zonal mean profiles only, the authors are essentially fitting a single scenario (four individual signal scans). They must demonstrate how the rates derived from zonal mean profiles fit real single scans in measured emission bands. This will show whether the derived rates have any value for practical analysis of measurements of both instruments.

Technical corrections:

Line 26-28. This sentence needs a citation at the end.

Lines 535-539. These sentences should be moved to section 2.3: The OH airglow Base model.

---

## Author Comment (AC1) · 12 Nov 2018

**Response to Referee #1**

**We thank the reviewer for the useful suggestions to improve the paper. The comments of the referee are repeated in bold letters while our response is given in normal text.**

**According to the comments of both referees, we changed to title of the paper, replaced X by [X], and added error bars to the TIMED/SABER observations in Fig. 1-5.**
**We further carried out sensitivity runs with different sets of Einstein coefficients and included a new Figure 2. We also increased the uncertainty of the Einstein coefficients and added uncertainties of SABER temperature, SABER OH(9-7)+OH(8-6) VER, and SABER O3, resulting in larger total uncertainties of [O(3P)] and [H]. The discussion of potential error sources of [O(3P)] was also extended.**
**The rate of OH(v=8)+O(3P) was reduced in order to obtain physically allowed [O(3P)] values, which are slightly lower than in the previous paper version.**
**Finally, a detailed comparison between the [O(3P)] derived here and [O(3P)] from other studies is also included in the section "Conclusions" and we explicitly state that out [O(3P)] should be regarded as an upper limit.**

**General comments:**

**I know it's a bit persnickety, but throughout the paper you need to be careful distinguishing between X and [X], as is done in the equations. X is not being derived, you are deriving X densities, or deriving [X].**
Done, we changed X to [X] throughout the paper.

**If O3 is also a variable in the airglow model, could you not compare the resulting O3 with SABER values as a further constraint, in addition to the SABER VER? Either way, it would be interesting to see how the best fit model O3 compares to the SABER values, since those are not related to OH (although if it is expected that SABER O3 values are too large, maybe this wouldn't work. Or could you compare to SABER O3 1.27 µm data?).**
No, O3 is not a variable in this paper and was obtained from SABER observations at 9.6 µm. Comparisons with SABER O3 at 1.27 µm are not possible since these measurements are not available during night. Recent comparisons between SABER night-time O3 with MIPAS night-time O3 showed that these two data sets agree within the corresponding error bars in the altitude region 80-100 km over the equator region (Lopez-Puertas et al., 2018, their Fig. 8 and 10). Thus, at least to our knowledge there is no conclusive evidence stating that SABER night-time O3 is generally too large or too low. The corresponding sentences in the paper were rephrased and an uncertainty of about 10 % of SABER O3 (Smith et al., 2013) was considered when estimating the total error of derived [O(3P)] and [H] profiles.

Thus, we added (l. 166-167):
"There are also SABER $O_3$ measurements at 1.27 µm but these observations are not available during night."

rewrote l. 211-216:

"Finally, rewriting Eq. (1) enables the derivation of [H] while [O($^3$P)] is calculated by substituting Eq. (3) in Eq. (1) and rewriting the resulting term as follows:

$$[H] = \frac{OH(9-7) + OH(8-6)\ VER}{G\,k_1\,[O_3]} \quad , \tag{4a}$$

$$[O(^3P)] = \frac{OH(9-7) + OH(8-6)\ VER}{G\,(k_3\,[O_2][M] - k_2\,[O_3])} \ . \tag{4b}$$

Air temperature and air pressure from SABER were used to calculate [M], [O$_2$] (VMR of 0.21), and [N$_2$] (VMR of 0.78) as well as to convert SABER O$_3$ VMR into [O$_3$] via the ideal gas law."

and added l. 549-554:

"Recent comparisons between MIPAS O$_3$ and SABER O$_3$ derived at 9.6 µm were performed by Lopez-Puertas et al. (2018). The authors showed that night-time O$_3$ from SABER is slightly larger than night-time O$_3$ obtained from MIPAS in the altitude region 80-100 km over the equator (their Fig. 8 and 10) but these differences are within the corresponding errors. Thus, at least to our knowledge there is no conclusive evidence stating that SABER night-time O$_3$ is generally too large. Nevertheless, we considered an uncertainty of O$_3$ of about 10 % (Smith et al., 2013)."

**Also, please comment on how initial conditions of the target species affect the results of the model, i.e. have you tested this, what are the scale of any uncertainties the first guesses can add?**

The target species [O(3P)] and [H] were derived by Eq. 4a and 4b, solely depending on OH airglow, [O3], [O2], [M], and several rates of chemical and physical processes involved (k1, k2, k3, G).

$$[H] = \frac{OH(9-7) + OH(8-6)\ VER}{G\,k_1\,[O_3]} \quad , \tag{4a}$$

$$[O(^3P)] = \frac{OH(9-7) + OH(8-6)\ VER}{G\,(k_3\,[O_2][M] - k_2\,[O_3])} \ . \tag{4b}$$

During our sensitivity runs, we used different [O(3P)] and [H] values based on different assumptions of the chemical and physical rates involved.

But we did not assume any a priori information of [O(3P)] and [H] to calculate these two target species, and consequently there are no "initial conditions" of the target species [O(3P)] and [H] influencing the model results.

Thus, we rephrased (l. 220-222):

"It is apparent from Eq. (4a-b) that any changes applied to the input parameters (G, O$_2$, O$_3$, M, k$_1$, k$_2$, k$_3$) are balanced by the derived values of [O($^3$P)] and [H], without assuming any a priori information of [O($^3$P) and [H]."

**Specific comments:**

**Abstract should specifically indicate that the [O] and [H] profiles derived in this study are from the SABER observations using an OH model informed by SCIAMACHY and SABER observations.**

We rewrote the beginning of the Abstract as follows (l. 10-16):

"Based on the zero dimensional box model CAABA/MECCA-3.72f, an OH airglow model was developed to derive night-time number densities of atomic oxygen ($[O(^3P)]$) and atomic hydrogen ([H]) in the mesopause region (~75-100 km). The profiles of $[O(^3P)]$ and [H] were calculated from TIMED/SABER satellite OH airglow emissions measured at 2.0 µm. The two target species were used to initialize the OH airglow model, which was empirically adjusted to fit four different OH airglow emissions observed by the satellite/instrument configuration TIMED/SABER at 2.0 µm and at 1.6 µm as well as measurements by ENVISAT/SCIAMACHY of the transitions OH(6-2) and OH(3-1)."

**L27: "high" should be "large" (as to not confuse with altitude)**

Sentence was rephrased.

**L39 and onward: What is meant by "OH(v)"? Do you mean vibrationally excited OH? It should be defined when it is first used as "vibrationally excited OH" or "OH(v>0)".**

We rephrased l. 38-39 and adapted the text onward:

"This chemical reaction additionally leads to the production of vibrationally excited hydroxyl radicals (OH(v>0)) up to the vibrational level v=9, …"

**L63: "last decades" sounds ominous. Should be specific, i.e. last three to four decades.**

Done. Was changed as suggested by the referee.

**L69: "of" should be "from"**

Done.

**L79: "individually" doesn't sound right. Maybe, "Both airglow emissions were used to derive separate data sets of O(3P) profiles"?**
**and**
**L80: "profiles" makes it sound as if only one profile was retrieved for each airglow feature. Should probably be "data sets".**

Done. Sentence was changed to:

"Both airglow emissions were used to derive separate data sets of $[O(^3P)]$ and the best agreement between these two $[O(^3P)]$ data sets was obtained …"

**L89: should be OH(v=9). Or define that OH(x) means OH(v=x).**

Done, the notation of OH(x) was changed to OH(v=x) throughout the paper.

**L140: please fix the significant digit mismatch for "837.5-848"**
Done, was changed to: "837.5-848.0".

**L183: by "issues" do you mean uncertainties?**
We changed "issues" to "uncertainties".

**L205-208: should specify that the three-body reaction is the production of O3.**
Was changed to:
"In the second step, chemical equilibrium of $O_3$ during night is assumed as follows:
$$k_1[H][O_3] + k_2[O(^3P)][O_3] = k_3[O(^3P)][O_2][M], \qquad (3)$$
meaning that $O_3$ loss due to H and $O(^3P)$ (left side) is balanced by $O_3$ formation via the three-body-reaction $O(^3P)+O_2+M$ (right side)."

**L209: M is not the total density of air, M represents an air molecule. [M] is the total density.**
Was changed to:
"while $M$ being an air molecule and $[M]$ being the total number density of the air."

**L215-216: The wording makes it sound as if the SABER O3 was derived via the ideal gas law. Did you mean to say that you're using SABER derived O3?**
Yes, we meant that the O3 volume mixing ratios from SABER were converted into O3 number densities.

Thus, we rephrased this sentence to:
"Air temperature and air pressure from SABER were used to calculate $[M]$, $[O_2]$ (VMR of 0.21), and $[N_2]$ (VMR of 0.78) as well as to convert SABER $O_3$ VMR into $[O_3]$ via the ideal gas law."

**L234-235: would suggest "well suited" as opposed to "very suited".**
Done.

**L248-250: the way this sentence is worded means that the equation should be v=9. If that's not the case, it should read something like "OH at all vibrational levels v≤9"**
The sentence was rephrased to:
"The reaction $H+O_3$ can populate $OH(v)$ at all vibrational level $v \leq 9$ and the nascent distribution of $OH(v)$ was taken from Adler-Golden (1997)."

**L264: I assume that by "added" you mean "applied" and not literally added.**
Yes, you assumed right. Since a factor cannot be added, we replaced "added" with to "applied".

**Figures 1-5: Why are there no error bars on the SABER observations?**
We added error bars of OH(5-3)+OH(4-2) VER in Fig.1-5 and a short description
as follows (l. 176-181):

"The total uncertainty of SABER OH airglow data used here comprises three different error sources. Since we used climatology of the measurements (see Sect. 2.2), there are sufficient samples that the random noise component of the total uncertainty is essentially zero. The remaining two major terms are the absolute calibration error ($<5$ %) and the "unfilter" factor error ($<3$ %). Assuming a root-sum-square propagation of the individual uncertainties, this results in a total uncertainty of about 6 % for all data points presented in this study."

**L396: "probably" is not needed**
Was deleted.

**L402: They also seem to match within the error bars above ~92 km.**
**and**
**L402-404: I believe this sentence is missing an altitude value and a very important comma. Are you intending to say, "The model still overestimates the measurements in the altitude region above xx km, which might be related to O(3P) quenching."?**
This section was rephrased as follows (l. 400-404):
"This new model is referred to as "$O_2$ SD model" and the corresponding results are displayed in Fig. 3 as red lines, showing that the simulated OH(6-2) VER matches the observations within the error bars below 85 km and above ~92 km. The model still overestimates the measurements in the altitude region ~90 km, which might be related to O($^3$P) quenching (see Sect. 3.3)."

**R8: this claims that you're only considering $0 \leq v' \leq v-5$, for $v \geq 6$. If that were the case, then the branching ratios for 8-4, and 7-3 should be 0, which, according to Table 2, they are not. Should it be $0 \leq v' \leq v-4$?**
This is a typing error. It has to be "$v' \leq 5$" and not "$v' \leq v-5$".
Thus, we corrected R8 to: OH($v \geq 6$)+$O_2 \rightarrow$ OH($v' \leq 5$)+$O_2$

**L432: "Including R10b in the model…" is confusing.**
**In the v-4 scenario, are you including R10a and R10b, or are you including only R10b and not R10a. If it's the former, that would seem to imply that v'=v-4 can't occur at all (for $v \geq 6$), and then, again, the branching ratios for 8-4, and 7-3 should be 0. If it's the latter, then I agree that the implication is that v'=v-5 (and not v'=v-4) is the predominant pathway, which fits with the values in Table 2. Please make the explanation of this case clearer. (It's even more confusing in the context of R8, which already says this pathway isn't being considered.)**
We meant the latter case, in which R10a is substituted by R10b.
Thus, we rephrased the sentence to: "Replacing R10a by R10b in the model…"

**L460: Should be "that implied" instead of "which implied".**
**Also, "implied" is somewhat vague and makes it sound like you might not be sure (same with "seems reasonable").**
Done. We changed "which implied" to "that showed" and "seems reasonable" to "is reasonable".

**Table 3: Reactions 11a-d seem to indicate that k11 doesn't entirely decrease with v, which goes against what's written in the text. This is touched on a bit later, but not explicitly stated.**

The rate of OH(v=8)+O($^3$P) was reduced and the corresponding explanation in the text was extended as follows (l. 473-486):

"The assumption that $k_{11}$(v) decreases at lower vibrational levels is supported by the overall rate of OH(v=7)+O($^3$P)→OH(v')+O($^1$D) at mesopause temperature which is suggested to be on the order of 0.9-1.6×10$^{-10}$ cm$^3$ s$^{-1}$ (Thiebaud et al., 2010; Varandas, 2004). At least to our knowledge, the total rate of OH(v=8)+O($^3$P)→OH(v')+O($^1$D) was not measured. Nevertheless, results reported by Mlynczak et al. (2018) and Panka et al. (2017, 2018) indicate that this rate might be slower than the value of 2.3×10$^{-10}$ cm$^3$ s$^{-1}$ suggested by Sharma et al. (2015). This is also in agreement with our findings here, because applying 2.3×10$^{-10}$ cm$^3$ s$^{-1}$ for $k_{11}$(v=9,8) results in non-physical [O($^3$P)] values above 90 km. The corresponding value of [O($^3$P)] e.g. at 95 km is about 1.25 times larger than SABER [O($^3$P)] 2013 (Mlynczak et al., 2013a) which in turn is about 1.15 times larger than the upper limit of [O($^3$P)] (Mlynczak et al., 2013b, their Fig. 4). This results in a factor of 1.15×1.25=1.44 (=44 %) above the upper limit and cannot be explained by the uncertainty of the [O($^3$P)] profile derived here (40 %, see Sect. 3.4). In order to obtain reasonable [O($^3$P)] values, it was necessary to lower the rate of $k_{11}$(v=8) to 1.8×10$^{-10}$ cm$^3$ s$^{-1}$, and we therefore recommend $k_{11}$(v=8)≤1.8×10$^{-10}$ cm$^3$ s$^{-1}$ as an upper limit to derive physically allowed [O($^3$P)] values."

**L528: "higher" should be "larger" as not to be confused with the discussion of altitude.**
Done.

**Figure 6: These plots would be much easier to read with boxed axes (ticks on the top and right).**
Done.

**Also, this would be a good spot to compare O3 and show that the model O3 is (presumably) lesser than SABER values.**
As explained above, O3 was not a variable in this study. The [O3] used here was calculated from SABER O3 VMR.

**L632-635: Have you considered doing a similar study incorporating OH(9-4), (8-3), and (5-1) band VERs from OSIRIS?**
Not yet because our project is focused on SCIAMACHY observations. Comparisons between SCIAMACHY and OSIRIS might be hard since both instruments are on board of two different sun-synchronous satellites. The Odin satellite crosses the equator at 18 LT while ENVISAT crosses the equator at 22 LT. Thus, there might be a few co-location measurements but only at high latitudes. But replacing SCIAMACHY data by OSIRIS data should be possible and it would be very useful to compare the corresponding results to the results of this study.

Thus, we added in the text:
"Including additional OH transitions, like OH(9-4), OH(8-3), and OH(5-1) from the Optical Spectrograph and InfraRed Imager System (OSIRIS) on board the Odin satellite, might result in other values and deactivation schemes. This could be a subject of a future study."

**Summary: needs a bit more description at the end of how [O] and [H] compare to the SABER results and explaining the differences.**

We added (l.653-663):

"The [H] derived here is systematically larger by a factor of 1.5 than SABER [H] reported in Mlynczak et al. (2018) which is primarily attributed to their slower $OH(\nu=8)+O_2$ rate. Our $[O(^3P)]$ values in the altitude region below ~87 km are in agreement within the corresponding errors with the results found in Mlynczak et al. (2018) and Zhu and Kaufmann (2018) but are lower than the values presented in Panka et al. (2018). However, we think that the results of the latter study are too large because the authors falsely assumed too fast $OH(\nu)+O_2$ rates. In the altitude region above ~87 km, the $[O(^3P)]$ shown here is generally larger than the values reported in these three studies up to a factor 1.5 to 1.7. These differences are attributed to the faster rates and different deactivation channels of $OH(\nu)+O(^3P)$. Therefore, it is indicated that we might overestimate $[O(^3P)]$ above >87km and we suggest that our results should be interpreted as an upper limit. However, a final conclusion cannot be drawn at this point due the large uncertainties of the rates assumed to derive $[O(^3P)]$."

---

## Author Comment (AC2) · 12 Nov 2018

**Response to Referee #2**

We thank the reviewer for the useful suggestions to improve the paper. The comments of the referee are repeated in bold letters while our response is given in normal text.

**According to the comments of both referees, we changed to title of the paper, replaced X by [X], and added error bars to the TIMED/SABER observations in Fig. 1-5.**
**We further carried out sensitivity runs with different sets of Einstein coefficients and included a new Figure 2. We also increased the uncertainty of the Einstein coefficients and added uncertainties of SABER temperature, SABER OH(9-7)+OH(8-6) VER, and SABER O3, resulting in larger total uncertainties of [O(3P)] and [H]. The discussion of potential error sources of [O(3P)] was also extended.**
**The rate of OH(v=8)+O(3P) was reduced in order to obtain physically allowed [O(3P)] values, which are slightly lower than in the previous paper version.**
**Finally, a detailed comparison between the [O(3P)] derived here and [O(3P)] from other studies is also included in the section "Conclusions" and we explicitly state that out [O(3P)] should be regarded as an upper limit.**

**General comments:**

**This study proposes a new OH airglow model to retrieve O and H densities in the mesosphere. The OH model is empirically developed to simultaneously fit four OH emissions observed by the SABER/TIMED instrument at 2.0- and 1.6-microns as well as the OH(6-2) and OH(3-1) bands measured by SCIAMACHY/ENVISAT. The authors show that using adjusted rate coefficients and specific state-to-state relaxation mechanisms, the OH model reproduces the four emissions. However, they retrieve very high O and H concentrations.**
**The concept of fitting four emission bands simultaneously is promising as it may constrain unknown parameters involved in modeling OH emissions. The conclusions regarding their new OH model, however, are speculative as model inputs used in the study to simulate emissions have larger uncertainties than the authors claim (i.e. Einstein coefficients, ozone concentration). Accounting for these uncertainties will significantly alter the results of this paper.**

**Further, the authors show that the applied OH model retrieves unrealistically high atomic oxygen and hydrogen in the MLT. Recent publications (s.f. Kaufmann et al. 2014, Mlynczak et al. 2018, Panka et al. 2018, and Zhu and Kaufmann 2018) have shown that [O] densities retrieved using SABER and SCIAMACHY measurements are much lower (by up to a factor 2 and more) than those retrieved in this study, specifically from 85-100 km. The model development (and the rate coefficient adjustments) must have a goal to reliably retrieve atmospheric properties from the observations. The very high [O] and [H] retrieved with the help of the new model indicates that there are still major flaws (it does not matter that it fits all selected emissions, this system has a very large number of unconstrained variables). The paper needs major revisions before making physical sense and being suitable for publication.**
We lowered the total rate of OH(v=8)+O(3P) and checked the validity of our [O(3P)]. Taking into account the uncertainty of O(3P) derived here, our [O(3P)] results are now physically justified between 80 km and 95 km based on radiative and energetic constraints reported in Mlynczak et al. (2013b). The estimation of total [O(3P)] uncertainty was extended by additionally including uncertainty of SABER

O3 (~10 %; Smith et al., 2013), SABER temperature (3 %, Garcia-Comas et al., 2008) and SABER OH airglow emissions (6 %).

We also carried out sensitivity runs with different data sets of Einstein coefficients to test their impact on our results. As a consequence, the uncertainty of Einstein coefficients was increased from 10 % to 30 %.

We added an extended comparison of our model results with recent results presented by Mlynczak et al. (2018), Panka et al. (2018), and Zhu and Kaufmann (2018). We further explicitly stated that our [O(3P)] should be viewed as an upper limit in the altitude region above 87 km. However, despite that our [O(3P)] results are larger than the [O(3P)] in these three studies, our results are physically allowed, while conditions of chemical equilibrium of O3 are also valid. Additionally, we did not claim that our results are undeniable truth. But we think that our study makes justified assumptions which is still of value for publication.

A detailed response is presented below, when answering your specific comments.

**Instead of fitting four emission bands while simultaneously retrieving [O] and [H] densities, I recommend for the revised study, to concentrate on retrieving [O] and [H] densities but only fit three emission bands (as will be discussed below, the OH(6-2) emission band is unreliable and taking into account its large uncertainty will alter the results of the current study).**

We agree that SCIMACHY OH(6-2) VER is relatively noisy. But this issue was taken into account by the relatively large error bars of these measurements and even these relatively large uncertainties cannot explain all the differences of OH(6-2) VER between the Base model and SCIAMACHY measurements. Additionally, the major impact of OH(6-2) VER on our model results is the suggestion that OH(v≥7)+O2 primarily contribute to OH(v≤5)+O2.

However, the corresponding changes in the OH model do account for an increase of [O(3P)] and [H] of about 10 %. And the impact on the derived [O(3P)] profile also decreases with increasing altitude. The large [O(3P)] values above 90 km are primarily caused by different OH(v)+O3P rates and the assumed deactivation paths. Thus, we kept SCIAMACCHY OH(6-2) VER observations in our study.

**Further, the authors must demonstrate how the rates derived from zonal mean profiles fit real single scans in three emission bands.**

This was done in Sect. 3.4 (l. 593-618) where we analyzed the relation between [O($^3$P)] and OH(9-7)+OH(8-6) VER presented in Fig. 7.

**Specific comments:**

**Line 1.**
**The title states "New insights in OH airglow modelling…". The proposed new "insights" are highly speculative and are inconsistent with existing theory and experiments. The authors need to first show that reliable [O] and [H] can be derived when their OH model is applied before claiming any new insights.**

The title was changed to: "Model results of OH airglow considering four different wavelength regions to derive night-time atomic oxygen and atomic hydrogen in the mesopause region"

**In the introduction section, discussion regarding the current progress of [O] and [H] retrievals using SABER and SCIAMACHY instruments is missing. Retrieving these two parameters are a key point of this study and no background is given. Please cite recent [O] and [H] retrieval studies and their proper discussion.**

We added (l. 100-117):

"The newly suggested rates of OH(ν)+O($^3$P) were applied in different models to derive [O($^3$P)] in the mesopause region. Mlynczak et al. (2018) used SABER OH airglow emissions observed at 2.0 µm to derive [O($^3$P)] and assumed rates of $3.0\times10^{-10}$ cm$^3$ s$^{-1}$ and $1.5\times10^{-10}$ cm$^3$ s$^{-1}$ for OH(ν=9)+O($^3$P) and OH(ν=8)+O($^3$P), respectively. They further stated that deactivation of OH(ν=9)+O($^3$P) has to occur via single-quantum quenching and that the OH(ν=8)+O$_2$ rate has to be smaller than known from laboratory measurements to get global annual energy budget into near balance. Panka et al. (2018) simultaneously investigated SABER OH airglow emissions measured at 2.0 µm and 1.6 µm, while applying faster rates for OH(ν=8)+O($^3$P) and OH(ν)+O$_2$. Their [O($^3$P)] values agree within the corresponding errors with the results reported by Mlynczak et al. (2018) above ~87 km but are larger in the altitude region below. The authors also demonstrated the high sensitivity of the derived [O($^3$P)] from O($^3$P) quenching rates applied in their model. Zhu and Kaufmann (2018) analyzed SCIAMACHY OH(9-6) transition. They used a value of $2.3\times10^{-10}$ cm$^3$ s$^{-1}$ for OH(ν=9)+O($^3$P) which is lower than the one applied in the two previous studies, resulting in generally lower [O($^3$P)] values in the altitude region above 87 km. Their rate for OH(ν=9)+O$_2$ lies between the corresponding rates of the two other studies, and consequently their [O($^3$P)] is also between the [O($^3$P)] values of these two studies below 87 km. Thus, recent publications indicate that the rate of OH(ν=9,8)+O($^3$P) might be slower than previously suggested in Sharma et al. (2015). But this problem needs further attention because all three papers derive different [O($^3$P)], depending on the data sets investigated."

**Lines 205-216.**
**The retrieval of [O] and [H] are both dependent on the [O3] volume mixing ratio. The authors used nighttime [O3] taken from SABER.**
**The nighttime SABER [O3] has never been rigorously validated, nor is there a paper discussing its retrieval approach. Differences between WACCM and SABER [O3] are roughly a factor of 2 (Smith et al. 2014) and, therefore, one cannot rely on SABER [O3] as an input parameter.**
**Additionally, in Mlynczak et al. 2018, the conclusion is made that current SABER daytime [O3] and, supposedly, the nighttime one is too high based on a significantly lower [O] retrieved in that study. It is clear from equation 4a and 4b that any variation in [O3] will have a significant effect on [O] and [H]. For the revised study, I recommend using inputs taken from a self-consistent photochemical model like WACCM instead of ones taken from retrievals, which are not supported by any other studies. Additionally, uncertainties in the retrieved parameters due to large uncertainties in the [O3] must be estimated and discussed.**

It is correct that chemistry-climate models like WACCM optimally contain descriptions of the state-of-the art of all known processes and usually provide a quite realistic representation of reality. However, these model results do not describe the "true" state of the atmosphere at any given point in time and space. Consequently, a comparison of models results and observations might be used to validate a model, but certainly not the observations.

In particular, WACCM has a well-known deficit of odd-oxygen, so it is not surprising that the WACCM O3 is less than SABER O3 or O3 obtained from any other measurement. This is a long-standing issue with WACCM and its predecessors. The daytime SABER O3 excess is certainly clear between 60 and 80 km. It may also be above that, but both Mlynczak et al. (2018) and Smith et al. (2014) could not conclude that above 80 km the daytime SABER O3 is too large.

The same is true for SABER night-time O3. Mlynczak et al. (2018) state that since "the cause of the larger daytime SABER ozone is not known, it is also possible that the SABER night O3 is also too large." This means that they cannot exclude an overestimation of their SABER night-time O3 because they do not know the reason behind SABER O3 daytime enhancement. But SABER night-time O3 might not have the same problem as SABER daytime O3. There are two candidates for the daytime O3 enhancement: One is the interfering bands of daytime $CO_2$ that are not properly accounted for. Secondly, there might be an out-of-band light leak in the spectral filter. Even if it is an out of band leak, it may not be an issue at night-time.

Furthermore, recent comparison between MIPAS O3 and SABER O3 (Lopez-Puertas et al., 2018) did not show conclusive evidence that SABER night-time O3 is generally too large between 80 km and 100 km.

Thus, we think SABER night-time O3 inside the time-space interval of interest is not less reliable than any other data set and we did not replace SABER O3. The corresponding sentences in the paper were rephrased and an uncertainty of 10 % of SABER O3 (Smith et al., 2013) was assumed and included in the calculation of the total [O(3P)] uncertainty.

We added in the paper (l. 549-556):
"Recent comparisons between MIPAS $O_3$ and SABER $O_3$ derived at 9.6 µm were performed by Lopez-Puertas et al. (2018). The authors showed that night-time $O_3$ from SABER is slightly larger than night-time $O_3$ obtained from MIPAS in the altitude region 80-100 km over the equator (their Fig. 8 and 10) but these differences are within the corresponding errors. Thus, at least to our knowledge there is no conclusive evidence stating that SABER night-time $O_3$ is generally too large. Nevertheless, we considered an uncertainty of $O_3$ of about 10 % (Smith et al., 2013). The uncertainty of SABER temperature was estimated to be lower than 3 % (Garcia-Comas et al., 2008) while the total uncertainty of SABER OH(9-7)+OH(8-6) VER was assumed to be about 6 % (see Sect. 2.1.2)."

**Lines 296-327.**
**"…we exclude the Einstein coefficients as a potential fundamental error source."**
**I do not agree with this statement. The new constraint imposed using the OH(6-2) emission band is unreliable. This band has a very small Einstein coefficient. The authors do not go into detail regarding the numerical differences among the literature of the OH(6-2) emission rate, but state that they are consistent.**
**The authors use the OH(6-2) coefficient taken from Xu et al. 2012 which is 1.767 sec-1. A more recent publication by Brooke et al. (2015) recalculated OH Einstein coefficients and found a rate of 1.16 sec-1 for the same transition. The rate of Xu et al. (2012) is approximately 50% larger than that of Brooke et al. (2015) and would significantly change the OH(6-2) emission profiles in Figures 1-5 as well as the results in Tables 2 and 3. The ab initio calculations of van der Loo and Groenenboom (2007, 2008) give values that are even smaller than Brooke et al. (2015) - the OH(6-2) emission rate of Xu et al. (2012) is 75% larger than that of van der Loo and Groenenboom (2007, 2008). Evidently, the issue of the OH Einstein coefficients is not yet settled.**
The problem of all these data sets of Einstein coefficients is that the results strongly depend on how good the representation of the Hamiltonian for the OH molecule is which is used to solve the Schrödinger equation. It is expected that the calculations improve with time, but not necessarily at these large quanta changes. Multi quanta transitions of more than 2 quanta have small Einstein coefficients and are generally hard to model and calculate.
Also, it is inappropriate solely focusing on the Einstein coefficient of OH(6-2) because errors of this single transition might be partly compensated by errors of other OH transitions. However, we agree

with the referee that different Einstein coefficient data sets have to be taken into account before excluding them as a potential error source.

Thus, we carried out sensitivity runs and the results are displayed in Fig. 2 (see next page). We also rephrased and extended the corresponding section in the text as follows:
"Since the overestimation of the Base model is especially large for OH(6-2) VER, an impact of the Einstein coefficient of the corresponding transition must be considered. Regarding this aspect, we have to point out that studies based on HITRAN 2004 data set should be viewed more critically, because of erroneous OH transition probabilities. The Einstein coefficients used in this study were recently recalculated (Xu et al., 2012, their Table A1) and correspond to a temperature of 200 K, which is very close to mesopause temperature. Furthermore, these Einstein coefficients are consistent with the values of the HITRAN 2008 data set (Rothman et al., 2009). However, there are several other data sets of Einstein coefficients found in literature that might lead to different results. We therefore carried out sensitivity runs, using the Einstein coefficients reported by Turnbull and Lowe (1989), Nelson et al. (1990), van der Loo and Groenenboom (2007), Xu et al. (2012; =Base model), and Brooke et al. (2016). The corresponding results are presented in Figure 2 and show considerably large differences in case of OH(6-2) VER which are about a factor of 4 between the highest and lowest model output. In contrast, the individual simulations of OH(5-3)+OH(4-2) VER and OH(3-1) VER are rather consistent and vary only by ~10 %. These results emphasize that the choice of the Einstein coefficients is a potential error source for higher quanta transitions.
Regarding the credibility of the Einstein coefficients, it is generally assumed that the calculation improve with time. However, this is not necessarily true at quanta changes >2 because it all depends on how good the representation of the Hamiltonian for the OH molecule is, that is used to solve the Schrödinger equation. Multi quanta transitions >2 quanta have small Einstein coefficients and are generally hard to model and calculate. The assessment of the Einstein coefficients requires a detailed analysis of the corresponding calculations, which is beyond the scope of this study. We therefore cannot exclude the values used in the Base model as a potential error source, but we also think that our choice of the Einstein coefficients from Xu et al. (2012) is reasonable. Additionally, these values represent approximately the average model output of all five data sets considered here, while the model results based on Nelson et al. (1990) and van der Loo and Groenenboom (2007) represent the variability. Thus, we will not replace the Einstein coefficients by Xu et al. (2012) in our model but keep in mind that they might be too large.
Furthermore, the best agreement between the observations and the model was obtained by applying the Einstein coefficients reported by van der Loo and Groenenboom (2007). But even in this case, the model still overestimates the observations of all OH transitions in the altitude region between ~80 km and ~86 km. This pattern strongly supports the suggestion stated above that the rates and schemes associated with OH($v$)+$O_2$ are incorrect."

We further added in the Conclusions (l. 635-637):
"Also note that the Einstein coefficients used here might be in error (see Sect.3.1; Fig. 2). This does not affect the two general conclusions drawn above but would impact the empirically derived rates."

[Figure]

**Figure 2 :** Same as Figure 1 but for different sets of Einstein coefficients from literature, namely N90 (Nelson et al., 1990), TL89 (Turnbull and Lowe, 1989), X12 (=Base model; Xu et al., 2012), B16 (Brooke et al., 2016), and vdLG07 (van der Loo and Groenenboom, 2007).

To make matters worse, the SCIAMACHY OH(6-2) band displays a signal count two orders of magnitude smaller than that of the OH(3-1) band. The uncertainty of the VER signals for these two bands will be vastly different. Finally, because these two bands have a different Delta v and values of Einstein coefficients that are different by more than one order of magnitude, the uncertainty of the OH(6-2) band will be much larger.

Therefore, both the observed low VER signals and the large uncertainty in the value of the Einstein coefficient indicate knowledge of the OH(6-2) band is highly uncertain. As a result, the OH(6-2) emission band cannot be confidently used to constrain OH modeling parameters. For

**manuscript revisions, I recommend to redo this study using only three OH emission bands. An alternative would be to validate the intensity of the OH(6-2) band by comparison with the OH(6-3) profile, which should be within the capabilities of SCIAMACHY.**

We agree with the referee that SCIAMACHY OH(6-2) is relatively noisy but this was taken into account by the larger error bars presented in Fig. 1-5. These error estimates are based on the random noise at altitudes above the OH emission (l. 147-151) and we do not see any reason, why the OH(6-2) band should be omitted here because of low signal-to-noise ratio.

Additionally, including OH(6-2) does not considerably affect the [O(3P)] derived here (~10 %) and the impact also decreases with increasing altitude. However, based on the OH(6-2) emissions, we suggest that OH(ν≥7)+O2 primarily contribute to OH(ν≤5)+O2.

Thus, we did not exclude SCIAMACHY OH(6-2) from our study. Whether future studies will obtain similar results or strongly disagree with our suggestion cannot be known. But suggesting a new idea is not wrong even when based on relatively noisy data because these uncertainties were considered.

**Lines 574-592.**
**"Applying their suggested limit, we found that in our case chemical equilibrium of O3 is probably true only above 80 km."**
**Recent studies have shown that the [O] and [H] retrieval approach used in this study may be flawed (Belikovich et al., 2018; Kulikov et al., 2017, 2018) and can introduce additional uncertainties. The authors addressed these issues very briefly here, but this needed to be more rigorously discussed. To say just simply "probably true" is insufficient. Additionally, uncertainties of the final results related to a probable chemical equilibrium breakdown need to be estimated and discussed.**

This section was rearranged and extended as follows:

"The second aspect influencing the quality of the derived profiles is the assumption of chemical equilibrium of $O_3$, represented by Eq. (3). This issue was recently investigated by Kulikov et al. (2018), which carried out simulations with a 3-D chemical transport model and demonstrated that a wrongly assumed chemical equilibrium of $O_3$ may lead to considerable errors of derived [O($^3$P)] and [H]. In order to test the validity of chemical equilibrium of $O_3$ locally, the authors suggested that OH(9-7)+OH(8-6) VER has to exceed 10×G×B, with $B$ including several chemical reaction rates involving $O_x$ and $HO_x$ species. Note that this criterion requires simultaneously performed temperature and OH airglow measurements. Furthermore, this criterion is based on the assumption that the impact of atmospheric transport on chemical equilibrium of $O_3$ is negligible. Since our experiments fit these conditions, we applied their suggested limit and found that in our case chemical equilibrium of $O_3$ is valid above 80 km. We have to point out that the term "chemical equilibrium of $O_3$" refers to $O_3$ that does not deviate more than 10 % from $O_3$ in chemical equilibrium (Kulikov et al., 2018, their Eq. 2). Assuming that $O_3$ is always 10 % above or below $O_3$ in chemical equilibrium introduces an uncertainty of about 10 % at 80 km and 20 % at 95 km, additionally to the total uncertainty of [O($^3$P)] and [H] estimated above. However, such a worst case scenario is rather unlikely while it is more realistic that $O_3$ actually varies around its chemical equilibrium concentration. Thus, an over- and underestimation of derived [O($^3$P)] and [H] are assumed to compensate each other. Consequently, we conclude that the impact on the total uncertainty of [O($^3$P)] and [H] due to deviations from chemical equilibrium of $O_3$ is negligible, but only because the previously used criterion is valid."

**Lines 615-617.**
**"… we think that the O(3P) and H derived by the Best-fit model provides reasonable results between 80 and 95 km."**

**The [O] derived looks somewhat reasonable only below 87 km, but not above this altitude. At 95 km, the retrieved [O] is at least two times larger than Mlynczak et al. [2018] and more than a factor of 5 at 100 km. It has also been discussed in detail that high [O] will disrupt the energy balance in the MLT (Mlynczak et al. 2013, 2018) and influence temperature retrievals. If, in the revised study, the retrieved [O] and [H] remain high, then please demonstrate how it impacts the heating and cooling of MLT and discuss in detail possible ways to overcome the corresponding energy budget imbalance.**

We compared our [O($^3$P)] derived here with the maximum [O($^3$P)] physically allowed by radiative constrains (Mlynczak et al., 2013b) and had to adjust our model to derive lower [O($^3$P)] values. This was done by reducing $k_{11}(v=8)$ from $2.3 \times 10^{-10}$ cm$^3$ s$^{-1}$ to $1.8 \times 10^{-10}$ cm$^3$ s$^{-1}$. Now, our [O($^3$P)] matches the upper limit suggested by Mlynczak et al. (2013b) within the corresponding errors.

Thus, we added in the text (l. 474-486):

"At least to our knowledge, the total rate of OH(v=8)+O($^3$P)→OH(v')+O($^1$D) was not measured. Nevertheless, results reported by Mlynczak et al. (2018) and Panka et al. (2017, 2018) indicate that this rate might be slower than the value of $2.3 \times 10^{-10}$ cm$^3$ s$^{-1}$ suggested by Sharma et al. (2015). This is also in agreement with our findings here, because applying $2.3 \times 10^{-10}$ cm$^3$ s$^{-1}$ for $k_{11}(v=9,8)$ results in non-physical [O($^3$P)] values above 90 km. The corresponding value of [O($^3$P)] e.g. at 95 km is about 1.25 times larger than SABER [O($^3$P)] 2013 (Mlynczak et al., 2013a) which in turn is about 1.15 times larger than the upper limit of [O($^3$P)] (Mlynczak et al., 2013b, their Fig. 4). This results in a factor of $1.15 \times 1.25 = 1.44$ (=44 %) above the upper limit and cannot be explained by the uncertainty of the [O($^3$P)] profile derived here (40 %, see Sect. 3.4). In order to obtain reasonable [O($^3$P)] values, it was necessary to lower the rate of $k_{11}(v=8)$ to $1.8 \times 10^{-10}$ cm$^3$ s$^{-1}$, and we therefore recommend $k_{11}(v=8) \leq 1.8 \times 10^{-10}$ cm$^3$ s$^{-1}$ as an upper limit to derive physically allowed [O($^3$P)] values."

**Line 638-639.**
**"Furthermore, it cannot distinguish between OH(5) and OH(4) as a well as OH(9) and OH(8), and consequentially errors in OH(5) and OH(9) might be compensated by errors in OH(4) and OH(8) or vice versa".**
**This is a troubling statement as your main results in Table 3 (R11a, R11b, R11c, R11d, and R11g) involve these levels and describe rate coefficients for specific state-to-state reactions. This statement needs to be clarified. It sounds as if you treat OH(9)+OH(8) as a combined, single level as well as OH(5)+OH(4). Is this true? If you cannot distinguish between certain vibrational levels, then how can you determine rate coefficients for specific vibrational levels?**

Yes, the SABER OH airglow emissions are a sum of OH(9-7) VER and OH(8-6) VER as well as OH(5-3) VER and OH(4-2) VER. We therefore cannot distinguish between OH(9) and OH(8) as well as OH(5) and OH(4).

However, when analyzing OH(v=9)+O($^3$P)→OH(0≤v'≤v-5)+O($^1$D), we can still draw some conclusion because this reaction can deactivate OH(v=9) only to OH(v≤4) but not OH(v=5). Thus, even if we cannot distinguish between OH(v=5) and OH(v=4) we can estimate a branching ratio of OH(v=9)+O($^3$P) → OH(v=4)+O($^1$D) if the total rate of OH(v=9)+O($^3$P) → OH(v≤4)+O($^1$D) is known. Furthermore, the rates of the individual paths presented in Table 3 are only a suggestion and not the main result. The main result of Table 3 is that the total loss rates of R11 are indicated to be slower compared to the values suggested by Sharma et al. (2015). These total rates presented here are a simplified solution in accordance with the rare laboratory experiments available and the OH transition considered. But as stated in the conclusion (l. 632-634): "Including additional OH transitions, …, might result in other values and deactivation schemes.

We rearranged this section as follows (l. 638-646):

"Furthermore, our OH airglow model is based on the transitions OH(9-7)+OH(8-6), OH(6-2), OH(5-3)+OH(4-2), and OH(3-1) only. Therefore, our model does not provide any information of OH($v\leq2$). It further cannot distinguish between OH($v=5$) and OH($v=4$) as well as OH($v=9$) and OH($v=8$), respectively, and errors in OH($v=5$) and OH($v=9$) might be compensated by errors in OH($v=4$) and OH($v=8$) or vice versa. Consequently, the rates of the individual deactivation channels presented in Table 2 and Table 3 should be viewed as a suggestion only. But these issues will only be solved eventually when future laboratory experiments provide the corresponding OH($v$)+$O_2$ and OH($v$)+O($^3$P) relaxation rates and deactivation channels. Nevertheless, we have to emphasize that the shortcomings of our model do not affect the two main conclusions drawn in this study."

**Tables 2 3. It is not clear if the results in Table 2 and 3 describe the Best-Fit model discussed in the conclusion. Table 2 shows empirically determined branching ratios of the OH(v) + O2 reaction for only VER observations "below 85 km" while Table 3 shows the branching ratios of the OH(v) + O(3P) reaction for only VER observations "above 85 km". The lack of consistency adds confusion to the findings of this study. Please clarify this. Is there not a best-fit model for altitudes 80-100 km?**

The individual model steps were always fit to the entire altitude interval 80-100 km. But OH(v)+O2 quenching is more important below 85 km while OH(v)+O(3P) becomes dominant above 85 km. Therefore, these altitudes were added in the caption of the tables.

But since this caused confusion we deleted "below 85 km" and "above 85 km" in the caption of Table 2 and 3.

**Table 3.**
**The two most important processes (largest rate coefficients) estimated from the best fits are not energetically allowed! Processes R11a and R11c are highly endothermic processes by ~3000 cm-1 and 2000 cm-1, respectively.**

The processes OH($v=9$)+O(3P)→OH($v=4$)+O(1D) and OH($v=8$)+O(3P)→OH($v=3$)+O(1D) are not findings of this study. They were adapted from OH($v$)+O($^3$P)→OH($0\leq v'\leq v$-5)+O($^1$D) suggested by Sharma et al. (2015) and were also included in other OH airglow models (e.g. Panka et al., 2017, 2018). Therefore, for details about their credibility, we refer to Sharma et al. (2015, and references within).

**Additionally, the state-to-state rate coefficients in Table 3 for the OH+O(3P) reaction appear to be in contradiction with the findings of Kalogerakis et al. (2016), who measured a large rate coefficient attributed to the resonant reaction OH(9)+O(3P)→OH(3)+O(1D). These results are non-physical and must be revised.**

Kalogerakis et al. (2016) only reported that OH($v=9$)+O(3P)→OH($v=3$)+O(1D) is an important deactivation channel of OH($v=9$)+O(3P)→products. They did not provide any rate or branching ratio of the channel. To our understanding, "important" means "not negligible" but it does not mean "dominating". Thus, we assumed that OH($v=9$)+O(3P)→OH($v=3$)+O(1D) has to occur but this channel is not necessarily the fastest deactivation path of OH($v=9$)+O(3P).

This was stated in the text (l. 490-493):

"However, not much is known about the individual branching ratios of R11 except that $OH(v=9)+O(^3P)\rightarrow OH(v=3)+O(^1D)$ is an important deactivation channel but not necessarily the dominating one (Kalogerakis et al., 2016)."

**As stated above, it seems most likely that fitting the highly uncertain OH(6-2) signal that has large systematic errors have skewed the results of this paper. Removing this constraint may bring the revised OH model into better agreement with recent laboratory and modeling studies as well as retrieve reasonable [O] and [H].**

As we wrote above, we took into account that OH(6-2) VER is a relatively noisy signal by considering the relatively large error bars. Also, OH(6-2) does not considerably affect derived [O(3P)] and [H] values (about 10 %).

Furthermore, we have to emphasize that, at least to our knowledge, the rates and deactivation schemes applied in the OH model are not in conflict with ANY laboratory measurements but partly disagree with other model studies. However, the intention of this study was not to match other model studies but to review recent rates and deactivation schemes, and consequently provide [O(3P)] and [H] values based on justified assumptions.

**Figures 1-5.**
**Why are there no error bars displayed for the measured OH(5-3)+OH(4-2) VER emissions despite error bars displayed for the OH(6-2) and OH(3-1) emissions?**
We added error bars of OH(5-3)+OH(4-2) VER in Fig.1-5 and a short description
as follows (l. 176-181):

"The total uncertainty of SABER OH airglow data used here comprises three different error sources. Since we used climatology of the measurements (see Sect. 2.2), there are sufficient samples that the random noise component of the total uncertainty is essentially zero. The remaining two major terms are the absolute calibration error (<5 %) and the "unfilter" factor error (<3 %). Assuming a root-sum-square propagation of the individual uncertainties, this results in a total uncertainty of about 6 % for all data points presented in this study."

**In general, the concept of fitting the zonal mean profiles for three OH bands is questionable. Operating with zonal mean profiles only, the authors are essentially fitting a single scenario (four individual signal scans). They must demonstrate how the rates derived from zonal mean profiles fit real single scans in measured emission bands. This will show whether the derived rates have any value for practical analysis of measurements of both instruments.**

We used climatology instead of single scans because the individual scans are too noisy to derive any reliable rate coefficients. We are aware that deriving [O(3P)] and [H] based on zonal mean climatology instead of a scan-to-scan basis does introduce additional uncertainties. In particular, this approach fails when linearity between [O(3P)] and OH(9-7)+OH(8-6) VER breaks down.

But this issue was considered in Sect. 3.4 (l. 593-618; Fig. 7):

"The last problem lies in the fact that the approach used here (see Sect. 2.2) has to be applied to individual OH airglow profiles to derive [O($^3$P)] and [H] correctly. However, the individual scans of OH(6-2) were too noisy to analyze single profiles and we therefore used climatology for all input parameters. By investigating individual OH airglow profiles, we would derive individual [O($^3$P)] profiles and eventually average them to the mean [O($^3$P)] profile. While in our case, we directly derive the mean [O($^3$P)] profile. This makes no difference as long as the relation between OH airglow and

[O($^3$P)] is a linear one. But Eq. (4b) shows that the relation between [O($^3$P)] and OH(9-7)+OH(8-6) VER is only approximately linear because $G$ also depends on [O($^3$P)], as represented by the terms $C_v$ and $C_{vv}$. The linearity between OH(9-7)+OH(8-6) VER and [O($^3$P)] of an air parcel with a certain temperature and pressure is solely controlled by [O($^3$P)]×G. Note that [H] too is affected by this non-linearity issue since [H] depends on $G$ (Eq. (4a)). Thus, derived [H] values are only reliable as long as the derived [O($^3$P)], and as a consequence $G$, is not seriously in error.

In order to test the linearity, [O($^3$P)]×G was plotted as a function of [O($^3$P)] and the corresponding results for Best fit model at five different heights are presented in Fig. 7. It is seen that the relation between [O($^3$P)] and [O($^3$P)]×G or OH(9-7)+OH(8-6) VER, respectively, is linear for small values of [O($^3$P)], while a non-linear behaviour becomes more pronounced for larger values of [O($^3$P)]. Furthermore, the starting point of the behaviour is shifted to lower [O($^3$P)] values at higher altitudes. In order to estimate this threshold, we performed a visual analysis and determined an upper limit of [O($^3$P)] before non-linearity of [O($^3$P)]×G takes over. The approximated upper limits are added as dashed lines in Fig. 7. Finally, an [O($^3$P)] value at a certain altitude is assumed to be true if this value is below the corresponding upper limit of [O($^3$P)]. Otherwise, it should be viewed more critically. This was done for each altitude and we found that the [O($^3$P)] and [H] profiles presented in Fig. 6 are plausible in the altitude region <95 km. In combination with the estimation of chemical equilibrium of $O_3$ and the maximum of physically allowed [O($^3$P)], we think that the [O($^3$P)] and [H] derived by the Best fit model are reasonable results between 80 km and 95 km. Note that these altitude limits do not affect the results with respect to OH(v)+O$_2$ and OH(v)+O($^3$P) presented in the Sect. 3.2 and 3.3."

**Technical corrections:**

**Line 29-31.**
**This sentence needs a citation at the end.**
Done, we added Andrews et al. (1987) and Mlynczak and Solomon (1993) as follows:
"Atomic oxygen in its ground state (O(3P)) and atomic hydrogen (H) strongly influence the energy budget in the mesopause region (~75-100 km) during day and night (Mlynczak and Solomon, 1993), and consequently affect atmospheric air temperature, wind, and wave propagation (Andrews et al., 1987)."

**Lines 631-646.**
**These sentences should be moved to section 2.3: The OH airglow Base model.**
We rearranged this section as follows (l. 629-642):
"We have to stress that we performed an empirical model study and the total rates and deactivation channels suggested here heavily depend on the OH transitions considered. Including additional OH transitions, like OH(9-4), OH(8-3), and OH(5-1) from the Optical Spectrograph and InfraRed Imager System (OSIRIS) on board the Odin satellite, might result in other values and deactivation schemes. This could be a subject of a future study. Also note that the Einstein coefficients used here might be in error (see Sect.3.1; Fig. 2). This does not affect the two general conclusions drawn above but would impact the empirically derived rates.
Furthermore, our OH airglow model is based on the transitions OH(9-7)+OH(8-6), OH(6-2), OH(5-3)+OH(4-2), and OH(3-1) only. Therefore, our model does not provide any information of OH(v≤2). It further cannot distinguish between OH(v=5) and OH(v=4) as well as OH(v=9) and OH(v=8), respectively, and errors in OH(v=5) and OH(v=9) might be compensated by errors in OH(v=4) and OH(v=8) or vice versa. Consequently, the rates of the individual deactivation channels presented in

Table 2 and Table 3 should be viewed as a suggestion only. But these issues will only be solved eventually when future laboratory experiments provide the corresponding OH($\nu$)+$O_2$ and OH($\nu$)+O($^3$P) relaxation rates and deactivation channels. Nevertheless, we have to emphasize that the shortcomings of our model do not affect the two main conclusions drawn in this study."

and added a shorter paragraph in Section 2.3 (l. 269-273):
"As described in the previous section, the OH airglow model is adjusted to fit OH(9-7)+OH(8-6) VER, OH(6-2) VER, OH(5-3)+OH(4-2) VER, and OH(3-1) VER. Thus, the model cannot provide information about OH($\nu \leq 2$). It further treats OH($\nu$=9) and OH($\nu$=8) as well as OH($\nu$=5) and OH($\nu$=4) as a single level and the corresponding deactivation channels should be viewed more critically. "

---

## Author Response (AR2)

**Response to Referee #1**

**We again thank the reviewer for the latest suggestions to improve the paper. The comments of the referee are repeated in bold letters while our response is given in normal text.**

**Line 144: Replace "added" with "applied"**
Done, was changed as suggested.

**Lines 216-217: Converting O3 VMR to [O3] does not involve the ideal gas law. I think what you want to say is something more like, "Air temperature and air pressure from SABER were used to calculate [M], [O2] (VMR of 0.21), and [N2] (VMR of 0.78) via the ideal gas law, and [M] was then used to convert SABER O3 VMR into [O3]."**
Done, was changed as suggested.

**Line 250: "level" should be "levels"**
Done.

**Line 589: Again, it's my opinion that "above" and "below" are ambiguous when discussing profile values (can be confused with height instead of value). I'd suggest changing "always 10 % above or below O3 in chemical equilibrium" to something like "always within 10% of O3…" or "always 10% greater or lesser than O3…"**
We agree with your suggestion but simply missed this error. Was changed to: "greater or lesser than…".

**Line 595-596: please explicitly state the criterion**
Was rewritten as follows:"…, but only because the previously used criterion (OH(9-7)+OH(8-6) VER>10×G×B) is valid."

**Response to Referee #2**

We again thank the reviewer for the latest suggestions to improve the paper. The comments of the referee are repeated in bold letters while our response is given in normal text.

**General Comments:**
**1. Kalogerakis et al. (2016) only reported that OH(v=9)+O(3P)→OH(v=3)+O(1D) is an important deactivation channel of OH(v=9)+O(3P)→products. They did not provide any rate or branching ratio of the channel. To our understanding, "important" means "not negligible" but it does not mean "dominating". Thus, we assumed that OH(v=9)+O(3P)→OH(v=3)+O(1D) has to occur but this channel is not necessarily the fastest deactivation path of OH(v=9)+O(3P).**

**This was stated in the text (l. 491-494):**
**"However, not much is known about the individual branching ratios of R11 except that OH(v=9)+O(3P)→OH(v=3)+O(1D) is an important deactivation channel but not necessarily the dominating one (Kalogerakis et al., 2016)."**
**…**
**Furthermore, we have to emphasize that, at least to our knowledge, the rates and deactivation schemes applied in the OH model are not in conflict with ANY laboratory measurements but partly disagree with other model studies.**

**Kalogerakis et al. (2016) measured the rate coefficient of the OH(v=9)+O(3P)→OH(v=3)+O(1D) pathway. Accounting for mesospheric temperatures, the authors suggested a rate of (2.3+/-1)e-10 cm3/sec. Your results suggest, for the same pathway, a rate coefficient 0.2 * 2.3e-10 cm3/sec, which is 4.6e-9 cm3/sec, about 3 times lower than the lower bound suggested by Kalogerakis et al. (2016). Whether the reaction is the dominating one or not, the rate coefficient you use for this pathway still does not agree with laboratory results.**

**In general, many O retrieval studies which use OH models only need to consider OH(v=9) and/or OH(v=8) and therefore, state to state pathways coupling higher and lower levels do not need to be considered carefully. Because you are fitting four bands and require detailed description of OH(v=3-9), these state to state pathways become very important. Currently your OH model disagrees with laboratory measurements, so a sentence in the conclusion and abstract should be added as a disclaimer noting the differences.**
We thank the reviewer for highlighting this issue.

We extended the Abstract (l. 20-22):
"The results also provide general support of the recently proposed mechanism OH(v)+O($^3$P)→OH(0≤v'≤v-5)+O($^1$D) but suggest slower rates of OH(v=8,7,6,5)+O($^3$P), partly disagreeing with laboratory experiments.", changed Lines 494-495 to:
"However, not much is known about the individual branching ratios of R11 except that OH(v=9)+O($^3$P)→OH(v=3)+O($^1$D) is an important deactivation channel but not necessarily the dominating one (Kalogerakis et al., 2016). These authors suggested a rate of 2.3(±1.0)×10$^{-10}$ cm$^3$ s$^{-1}$ and noted that this rate might be slower due to the involvement of excited surfaces.", and added in the Conclusions (l. 647-649):
"In particular, the rate of OH(ν=9)+O($^3$P)→OH(ν=3)+O($^1$D is about 3 times slower than the lower limit reported by Kalogerakis et al. (2016)."

**2. "Figure 6 displays the vertical profiles of [O(3P)] and [H] obtained by the Best fit model in comparison with the results derived from SABER OH(9-7)+OH(8-6) VER only (Mlynczak et al., 2018). The [O(3P)] profiles seen in Fig. 6a agree below 85 km but the Best fit model shows gradually larger values in the altitude region above. These larger values are caused by the different deactivation rates and schemes of OH(v)+O(3P), agreeing with general pattern reported in Panka et al. (2018). We have to point out that other studies (e.g. von Savigny and Lednyts'kyy, 2013) ·observed a pronounced [O(3P)] maximum of about $8\times10^{11}$ cm$^{-3}$ at 95 km."**

**It is clear in Figure 6 that your O begins to gradually increase in the upper altitudes. In a recent paper by Zhu & Kaufmann et al (2018), the authors show that O retrievals by Mlynczak et al. (2018), Panka et al. (2018), and Zhu and Kaufmann (2018) all show O densities which peak around 95 km (see Figures 2 and 3). I am confused by the statement where you report that your O follows the general pattern of Panka et al. (2018) as this increasing pattern is not shown in Panka et al. (2018) or Zhu and Kaufmann (2018). Please remove this sentence or clarify your meaning.**
This statement focused on the altitude region below 95 km and referred to the general increase of [O(3P)] below 95 km.
Thus, we added (l. 531-533):

[revised manuscript text omitted]